# Thinker: Learning to Think Fast and Slow

**Stephen Chung**[*]
University of Cambridge

**Wenyu Du**[*]
The University of Hong Kong

**Jie Fu**
Shanghai AI Lab

## Abstract

Recent studies show that the reasoning capabilities of Large Language Models (LLMs) can be improved by applying Reinforcement Learning (RL) to question-answering (QA) tasks in areas such as math and coding. With a long context length, LLMs may learn to perform search, as indicated by the self-correction behavior observed in DeepSeek R1. However, this search behavior is often imprecise and lacks confidence, resulting in long, redundant responses and highlighting deficiencies in intuition and verification. Inspired by the Dual Process Theory in psychology, we introduce a simple modification to the QA task that includes four stages: *Fast Thinking*, where the LLM must answer within a strict token budget; *Verification*, where the model evaluates its initial response; *Slow Thinking*, where it refines the initial response with more deliberation; and *Summarization*, where it distills the refinement from the previous stage into precise steps. Our proposed task improves average accuracy from 25.6% to 27.3% for Qwen2.5-1.5B, and from 45.9% to 51.0% for DeepSeek-R1-Qwen-1.5B. Notably, for Qwen2.5-1.5B, the Fast Thinking mode alone achieves 25.2% accuracy using fewer than 1000 tokens, demonstrating substantial inference efficiency gains. These findings suggest that intuition and deliberative reasoning are distinct, complementary systems benefiting from targeted training. Additionally, we have open-sourced both the trained models and the source code.

## 1 Introduction

Multiple studies have shown that the reasoning capabilities of Large Language Models (LLMs) can be enhanced by applying Reinforcement Learning (RL) to question-answering (QA) tasks [1, 2, 3], demonstrating impressive mathematical and coding performance across benchmarks. With long context lengths, an interesting emergent behavior is self-correction within the chain-of-thought (CoT), where the LLM learns to perform search, such as verifying its steps, backtracking, and trying alternative paths.

However, it has been observed that this emergent search tends to be inefficient—the CoT is often long and redundant [4, 5]. For example, Deepseek R1's reasoning typically involves excessive backtracking and verification [1]. A likely cause is inefficient temporal credit assignment: for instance, in the GRPO algorithm used to train Deepseek R1, the entire generation sequence receives the same scalar advantage. That is, if the final answer is correct, the probability of the whole sequence is increased—regardless of which parts were actually useful. As a result, futile search paths and uncertain verifications are also rewarded, as long as the correct solution is eventually produced. Consequently, *intuition*—the ability to identify promising search paths rapidly—and *verification*—the ability to evaluate a search path confidently—are not *explicitly* trained and may therefore be underdeveloped.

A typical RL solution to this issue is to use more precise temporal credit assignment, such as incorporating a critic to compute a more accurate advantage for each token, as in PPO [6]. However,

---

[*]Equal contribution. Correspondence to: `mhc48@cam.ac.uk`.

39th Conference on Neural Information Processing Systems (NeurIPS 2025).

studies show that PPO performs similarly to GRPO [7, 2]—indicating that the critic may not be accurate enough for token-level credit assignment. Another approach is to use a lower discount rate or a shorter context length to encourage more efficient search; however, this may hinder the emergence of search behavior, as studies show that a long context length is necessary for strong performance [8, 9].

To address this dilemma, we draw inspiration from how human decision-making is modeled under Dual Process Theory [10]. According to this theory, humans possess two distinct but complementary cognitive systems: System 1, which operates quickly and intuitively based on heuristics but is prone to biases, and System 2, which is slower, more deliberate, and capable of reflective reasoning. Within this framework, a typical decision-making process unfolds as follows:

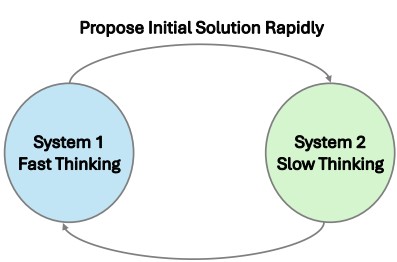

1. System 1 rapidly generates a candidate option based on intuition.

2. System 2 evaluates this option through mental simulation.

3. If the option passes verification, it is implemented; otherwise, System 2 attempts to refine it.

4. If refinement fails, the process returns to System 1 for another option.

Figure 1: Conceptual model of the interaction between Fast Thinking and Slow Thinking modes in the Thinker task, based on Dual Process Theory.

Inspired by this decision-making process, we propose the *Thinker task* as an alternative to the standard QA task. In a typical QA task, the model receives a question and generates a final answer in a single pass. A binary reward is given based solely on the correctness of the final answer. In contrast, the Thinker task decomposes the response into a four-step process:

1. **Fast Thinking**: The agent generates an initial answer using a small token budget.

2. **Verification**: The agent evaluates the correctness of the initial answer using a small token budget. If verified, it is accepted as the final answer.

3. **Slow Thinking**: If the initial answer fails verification, the agent can produce another final answer, using a large token budget.

4. **Summarization**: The agent summarizes the reasoning from the slow thinking step into a concise summary that leads to the same final answer.

We design distinct reward signals for each step, aiming to enhance different capabilities of the agent: intuition from Fast Thinking, evaluation from Verification, refinement from Slow Thinking, and integration from Summarization. Crucially, the reward signal for each task is restricted to that task alone. This separation allows for more precise temporal credit assignment by isolating learning signals for each task. For example, in the Fast Thinking task, the agent receives a binary reward based on the correctness of the initial answer, encouraging it to identify promising search paths under strict token budgets—thereby strengthening intuition. Meanwhile, the Slow Thinking task preserves the opportunity for the agent to learn a more general search strategy to refine previously incorrect answers.

The design facilitates a virtuous loop between intuition and reasoning. *Fast Thinking helps Slow Thinking by providing a promising initial search path, while Slow Thinking helps Fast Thinking by refining flawed intuition.* This bidirectional refinement mirrors how expert human decision-making evolves through repeated interactions between intuition in System 1 and reasoning in System 2 [11].

Experimental results validate our approach: relative to the QA task, the Thinker task yields consistent gains across diverse math benchmarks, with average relative performance gains of 6.7% for Qwen2.5-1.5B models and 11.1% for DeepSeek-R1-Distill-Qwen-1.5B models. Furthermore, our analysis reveals a notable reduction in reflection patterns, suggesting more direct reasoning. In summary, the proposed Thinker task offers the following key strengths:

- **Specialized Training**: Dedicated sub-tasks and rewards are designed to explicitly train distinct agent capabilities, providing richer and more targeted learning signals.

- **General Applicability**: The Thinker task can replace standard QA tasks without imposing constraints on the choice of RL algorithm or model architecture.

- **Inference Efficiency**: The Fast Thinking mode, requiring minimal token generation, can be deployed standalone for simpler tasks, offering a flexible trade-off between performance and computational cost during inference.

- **Strong Empirical Performance**: Our experiments demonstrate that agents trained with the Thinker task consistently outperform those trained on standard QA tasks across various benchmarks.

## 2 Background

In a single-turn QA task, a question is sampled from a dataset, and the LLM generates a response to the question. Concretely, let the dataset be denoted as $\mathcal{D} = \{(x_{(i)}, y_{(i)}^*)\}_{i=1}^N$, where $x_{(i)}$ denotes the $i$-th question, $y_{(i)}^*$ is its corresponding ground-truth answer, and $N$ is the size of the dataset. Let $\pi_\theta(\cdot \mid x)$ denote the model's policy, parameterized by $\theta$. A response $a \sim \pi_\theta(\cdot \mid x)$ is sampled for question $x$. The objective is to maximize:

$$J(\theta) = \mathbb{E}_{x, y^* \sim \mathcal{D}}[R(a, y^*)], \tag{1}$$

where $a \sim \pi_\theta(\cdot \mid x)$, and $R$ is the reward function, such as a binary function that returns 1 if the extracted answer from $a$ matches the ground-truth answer $y^*$, and 0 otherwise.

In a more general multi-turn task, we allow the dialogue to continue after the first response. Concretely, we denote $x_t$ and $a_t$ as the prompt and model response at turn $t$. The initial prompt $x_0$ is randomly sampled from the dataset $\mathcal{D}$. To generate the subsequent prompt $x_t$, we define the transition function $x_t = g(x_{0:t-1}, a_{t-1})$ (with $x_{a:b}$ endpoint-inclusive), which determines the next prompt based on previous prompts and responses, or whether to terminate the episode. Thus, the objective in the multi-turn task becomes:

$$J(\theta) = \mathbb{E}_{x_0, y^* \sim \mathcal{D}} \left[ \sum_{t=0}^T R_t(a_{0:t}, y^*) \right], \tag{2}$$

where $a_t \sim \pi_\theta(\cdot \mid x_{0:t}, a_{0:t-1})$, that is, the response is conditioned on all previous prompts and responses, and $T$ is the terminal step.

## 3 Method

In the proposed *Thinker* task, we decompose the QA task into four steps. The whole task occurs within a single dialogue, meaning that the agent receives all prompts and responses from previous steps in addition to the current prompt. An illustration of the Thinker task is shown in Figure 2.

### 3.1 Task Description

**Step 1 - Fast Thinking.** In the first step, the agent is prompted to answer the given question concisely within a strict token budget. The response in this step is restricted to a relatively short maximum generation length (e.g., 1000 tokens), meaning that the response will be truncated if it exceeds this length. The reward $R_{\text{fast}}$ in this step is defined as a binary function based on the correctness of the extracted answer. Specifically, let $y_{\text{fast}}$ denote the extracted answer; then $R_{\text{fast}} = 1\{y_{\text{fast}} = y^*\}$. The agent always proceeds to the next step after the response.

*Motivation.* The motivation of this step is to explicitly train the agent's intuition. As the agent must generate a response under a strict token budget, it cannot search extensively. It is usually restricted to a few search paths, which are directly reinforced if one leads to the correct answer.

**Step 2 - Verification.** In the second step, the agent is prompted to verify whether the fast answer $y_{\text{fast}}$ is correct, and must output either `Yes` or `No`. The response in this step is restricted to a relatively short maximum generation length (e.g., 2000 tokens). The reward $R_{\text{verify}}$ in this step is defined as a weighted binary function based on the correctness of the verification. Specifically, let $y_{\text{verify}}$ denote the extracted answer; then:

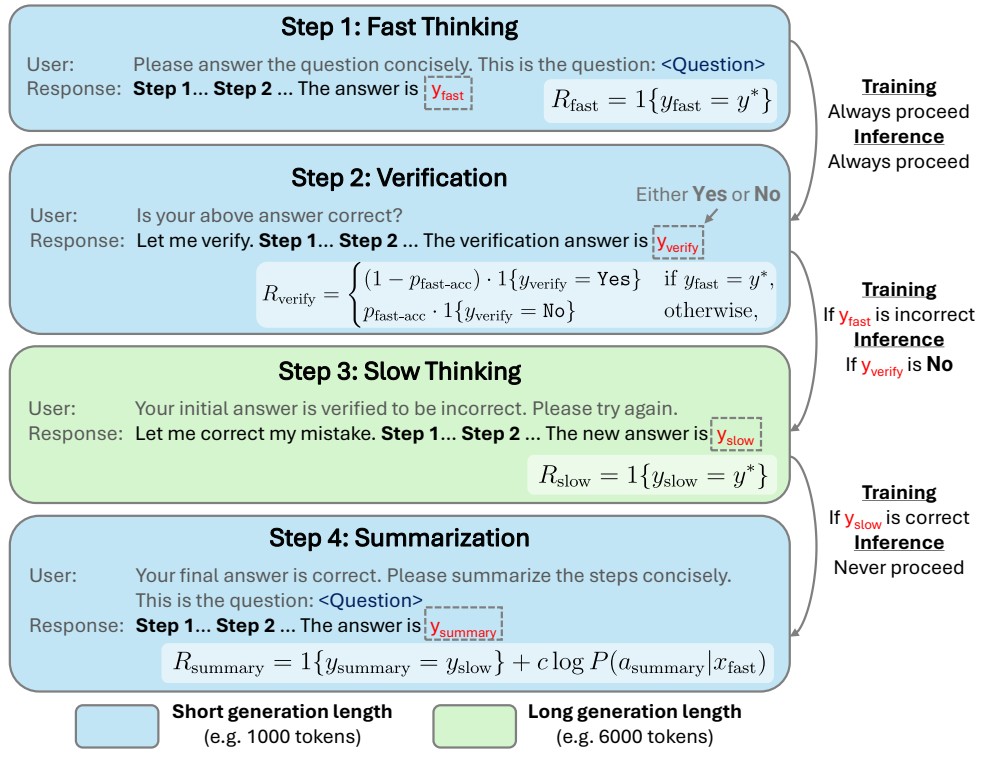

Figure 2: The four-step Thinker task. Each stage involves a user prompt, model response, and specific rewards and transition conditions designed to train distinct agent capabilities (intuition, evaluation, refinement, and integration). Reward function details are in the main text.

$$R_{\text{verify}} = \begin{cases} (1 - p_{\text{fast-acc}}) \cdot 1\{y_{\text{verify}} = \texttt{Yes}\} & \text{if } y_{\text{fast}} = y^*, \\ p_{\text{fast-acc}} \cdot 1\{y_{\text{verify}} = \texttt{No}\} & \text{otherwise}, \end{cases} \tag{3}$$

where $p_{\text{fast-acc}}$ denotes the trailing accuracy of the fast thinking step (averaged over the batch). This weighting is used to balance the two classes, so as to discourage the agent from always outputting `Yes` or `No` when the accuracy of the fast thinking step is too high or too low.

The transition function to the next step depends on whether the agent is in training or inference mode. In inference mode, if the agent answers `Yes`, then the fast answer is chosen as the final answer, and the episode terminates; otherwise, the agent proceeds to the next step. In training mode, if the fast answer is correct, then it is chosen as the final answer; otherwise, the agent proceeds to the next step. The distinction between training and inference mode aims to ensure that, during training, the Slow Thinking step primarily focuses on instances where the fast answer was incorrect. This prevents the agent from needing to re-verify an already correct fast answer in the Slow Thinking stage. However, during inference, we do not have access to the ground-truth answer, so we must rely on the agent's verification.

*Motivation.* The motivation of the second step is to explicitly train the agent's evaluation capability. The agent receives a clear binary reward based on whether its verification result is correct. Verifying an answer is often easier than generating one. If the fast answer is already verified to be correct, there is no need to proceed further, thus saving computational cost.

**Step 3 - Slow Thinking.** In the third step, the agent is prompted that the fast answer has been verified to be incorrect and is asked to try an alternative answer. The response in this step is restricted to a relatively long maximum generation length (e.g., 6000 tokens). The reward $R_{\text{slow}}$ in this step is defined as a binary function based on the correctness of the extracted answer. Specifically, let $y_{\text{slow}}$ denote the extracted answer; then $R_{\text{slow}} = 1\{y_{\text{slow}} = y^*\}$.

In both inference and training modes, $y_{\text{slow}}$ is always chosen as the final answer if the Slow Thinking step is executed. In inference mode, the episode ends here. In training mode, the agent proceeds to

the next step if the slow answer $y_{\text{slow}}$ is correct; otherwise, the episode ends. This distinction exists because the purpose of the next step—summarization—is to distill the long and correct response in this step to improve intuition, which is not applicable in inference mode.

*Motivation.* The motivation of this step is to encourage the agent to learn to refine incorrect fast answers for difficult questions. It should learn to use the reasoning from the verification step and revise errors to arrive at the correct answer. If such refinement is not possible, it should learn to try an alternative approach, leveraging the generous token budget for generation.

**Step 4 - Summarization.** In the fourth step, the agent is prompted that the previous slow answer is correct and is asked to concisely summarize the steps leading to it. Crucially, the prompt for the Summarization step includes the original question again, mirroring its presentation in the Fast Thinking prompt. The response in this step is restricted to a relatively short maximum generation length (e.g., 1000 tokens). The reward $R_{\text{summary}}$ in this step is designed based on two criteria:

1. **Correctness**: The extracted answer from the summary, $y_{\text{summary}}$, should be the same as the previous slow answer, meaning it should not produce a summary that leads to an incorrect answer.

2. **Consistency**: The response should be consistent with what the model is expected to produce in the Fast Thinking mode—that is, its probability conditioned on the Fast Thinking prompt should not be unduly low. For example, directly outputting the final answer without intermediate steps is considered inconsistent, as the likelihood of producing the correct answer directly under Fast Thinking mode is typically very low.

Combined, the reward function in this step is defined as:

$$R_{\text{summary}} = 1\{y_{\text{summary}} = y_{\text{slow}}\} + c \log P(a_{\text{summary}} \mid x_{\text{fast}}), \tag{4}$$

where $x_{\text{fast}}$ is the prompt in Fast Thinking step (i.e., the initial prompt), $\log P(a_{\text{summary}} \mid x_{\text{fast}})$ is the log probability of the summary response under the Fast Thinking prompt, and $c$ is a scalar hyperparameter. In experiments, we found that the agent sometimes still degenerates to give a very short answer despite the log probability term. To mitigate this, we gate the reward to $0$ if the length of the generated response is less than a low threshold (e.g., 300 tokens).

*Motivation.* The motivation of the final step is to reinforce concise reasoning patterns by rewarding correct and consistent summaries. A key design element is that the original question is re-presented in the Summarization prompt, mirroring its appearance in the Fast Thinking step. The agent is trained to produce a concise reasoning trace that leads to the correct answer for this input. This encourages the model to form a strong association between the original question and a correct, concise solution path. We hypothesize that this targeted reinforcement distills the successful but lengthy reasoning from Slow Thinking into a compact form suited to the Fast Thinking mode—thereby improving the agent's intuition. In addition to intuition, this step also trains the agent's integration ability, as it must extract and condense key reasoning steps from the longer trace generated in the previous step.

### 3.2 Training with the Thinker Task

Training LLMs with the Thinker task requires particular considerations regarding reward propagation. Since the reward at each step is specific to that step alone, it should not be propagated backward to earlier steps. This implies that the discount factor between steps should be set to $0$, while that within each step should be high (e.g., $1$) to enable effective credit assignment over tokens. The Thinker task defines only the RL environment and imposes no restrictions on the choice of algorithm or model architecture, allowing compatibility with any standard RL method (e.g., PPO) and LLM.

## 4  Related Work

**Environment Augmentation.** Our *Thinker task* is a form of environment augmentation for QA, inspired by Dual Process Theory and related to concepts like the Thinker MDP [12, 13]. While Thinker MDP provides agents with a simulated world model for interaction before action, our task structures QA into stages where self-generated intermediate outcomes guide subsequent reasoning. This contrasts with multi-attempt tasks [14] that allow iterative revision but require ground-truth access during inference, a constraint our method avoids.

**RL and Reasoning in LLMs.** A large number of studies have demonstrated the effectiveness of applying RL to enhance the reasoning capabilities of LLMs [1, 2, 3, 9]. Our work builds on these efforts by decomposing the QA task into the four-step Thinker task. It has been observed that LLMs trained with RL can produce inefficient CoT [4], leading to *overthinking* [15]. Our work relates to strategies for controlling generation length (often termed token budgets). Examples include dynamic token budgets that scale with problem complexity [5], or user-defined budgets [16, 17]. For instance, Muennighoff et al. [18] utilize token budget controls during the generation of CoT data for supervised fine-tuning (SFT). Concurrent works such as ThinkPrune [19], Concise Reasoning [20], SR-Flow [21], and AdaptThink [22] modify the reward or RL algorithm to encourage more efficient reasoning. While these approaches primarily focus on token budget control within a single response generation, our method introduces a structured, multi-step RL task explicitly designed to train distinct agent abilities independent of the underlying RL algorithm. In addition, our work is related to methods for encouraging self-correction in LLMs [23]. For instance, methods like Self-Refine [24] and Reflexion [25] primarily use prompt engineering and few-shot examples to enable agents to incorporate internal or external feedback for refining subsequent responses. However, unlike our work, these methods typically do not involve RL fine-tuning of the agent for these self-correction behaviors; instead, the correction capability is elicited at inference time through prompting.

# 5 Experiments

This section details the experiments conducted to assess whether the Thinker task can more effectively enhance LLM reasoning capabilities compared to a standard QA task. We focus on the mathematical reasoning domain here.

## 5.1 Experimental Setup

To evaluate the Thinker task, we fine-tune two publicly available models: Qwen2.5-1.5B (Q1.5B) [26] and DeepSeek-R1-Distill-Qwen-1.5B (R1.5B) [1]. While sharing a base architecture, R1.5B has undergone additional distillation using reasoning data from DeepSeek-R1, endowing it with stronger initial reasoning and search behaviors compared to the base Q1.5B. Training both models allows us to investigate the Thinker task's impact on a model with foundational capabilities (Q1.5B) and its ability to further enhance a model already specialized for reasoning (R1.5B). For the Q1.5B runs, we use three independent seeds and report the averaged results.

We fine-tune these models using RL on both the Thinker task and a standard QA task (serving as our baseline). For all experiments, we employ PPO [6]. Key hyperparameters include a discount rate $\gamma = 1$, GAE lambda $\lambda = 1$, and a sampling temperature of 1. No KL penalty against a reference policy was applied. The Fast Thinking and Summarization stages use a 1000-token budget, Verification uses a 2000-token budget, and Slow Thinking uses a 6000-token budget. Most other hyperparameters and training details mirror those in Open-Reasoner-Zero, with details provided in Appendix A.

For the training data, we utilized the 129K math question-answering dataset provided by Open-Reasoner-Zero. Each training run for both the Thinker task and the baseline required approximately 7 days on two compute nodes, each equipped with 8 A100 GPUs. Our implementation, adapted from the Open-Reasoner-Zero codebase [3], is publicly available at `https://github.com/stephen-chung-mh/thinker-task`, which also includes the trained models.

## 5.2 Training Dynamics and Evaluation Results

The training performance is shown in Figure 3, measured by accuracy on the training data. For the Thinker task, we plot fast accuracy (from the Fast Thinking stage) and final accuracy (from the Slow Thinking stage). Note that the final accuracy is not directly comparable to the baseline's, as the Thinker agent effectively has two attempts during training (though only one during testing). We observe that the Thinker agent's fast and final accuracy improve steadily, while the baseline's accuracy plateaus rather quickly. This may suggest that the Thinker agent is learning the underlying intuition of the task, which requires sustained training, as opposed to merely acquiring a generic search algorithm. We also observe that the fast accuracy of the Q1.5B model surpasses the baseline's. Notably, while

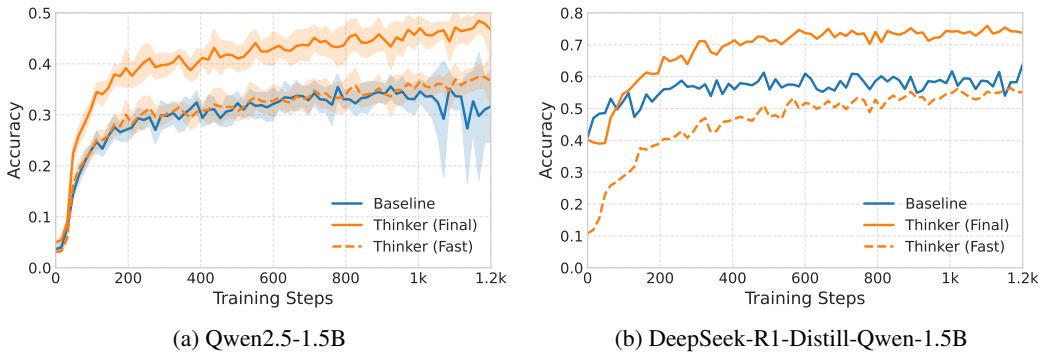

(a) Qwen2.5-1.5B

(b) DeepSeek-R1-Distill-Qwen-1.5B

Figure 3: Accuracy on training set. For the Qwen2.5-1.5B model (a), the shaded region represents the standard deviation across three independent seeds.

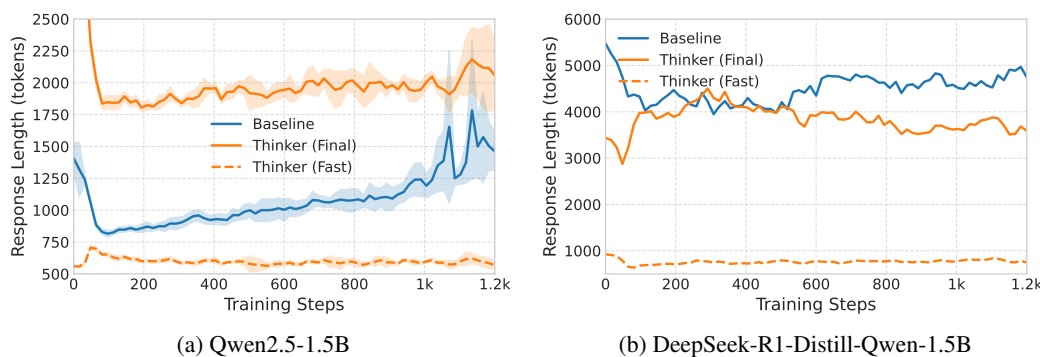

(a) Qwen2.5-1.5B

(b) DeepSeek-R1-Distill-Qwen-1.5B

Figure 4: Average response length on training set. For the Qwen2.5-1.5B model (a), the shaded region represents the standard deviation across three independent seeds.

both metrics reflect a single attempt, the Fast Thinking mode achieves superior performance with a significantly smaller token budget (Figure 4), underscoring its efficiency.

Figure 4 illustrates the average response length during training, plotting both the Fast Thinking response length and the cumulative length across all four stages of the Thinker task. The two models exhibit distinct behaviors. For the Q1.5B model, which lacks strong inherent self-correction capabilities, the Thinker task's cumulative length is greater than the baseline's, as its structured stages introduce verification and refinement steps. Conversely, for the R1.5B model, which already possesses self-correction abilities, the Thinker agent's total response length eventually becomes shorter than the baseline's. This suggests that rather than simply adding overhead, the Thinker task refines the R1.5B model's existing search process, pruning inefficiencies to achieve more token-efficient reasoning.

To evaluate the models, we consider the following common benchmarks: MATH500, AIME2024, AIME2025, GPQA Diamond, Olympiadbench, AMC23, and Minerva Math. We evaluate agent Pass@1 accuracy on these benchmarks every 50 training steps. The average performance across the benchmarks during training is shown in Figure 5. The trend is similar to the training performance, with the agent trained on the Thinker task surpassing the baseline. Detailed breakdowns for each benchmark are provided in Appendix B. Notably, for the R1.5B model, a moderate gap exists between its final accuracy and fast accuracy. This disparity is likely attributable to R1.5B's strong inherent reasoning capabilities, enabling its Slow Thinking mode to effectively utilize the larger token budget for refining initial answers.

The performance of the final models is detailed in Table 1. These models correspond to the check-points, saved at 50-step intervals, that achieved the highest accuracy on a validation dataset. We evaluated these final models on the aforementioned benchmarks and the additional CollegeMath benchmark; due to its large size, CollegeMath was reserved for this final evaluation and was not

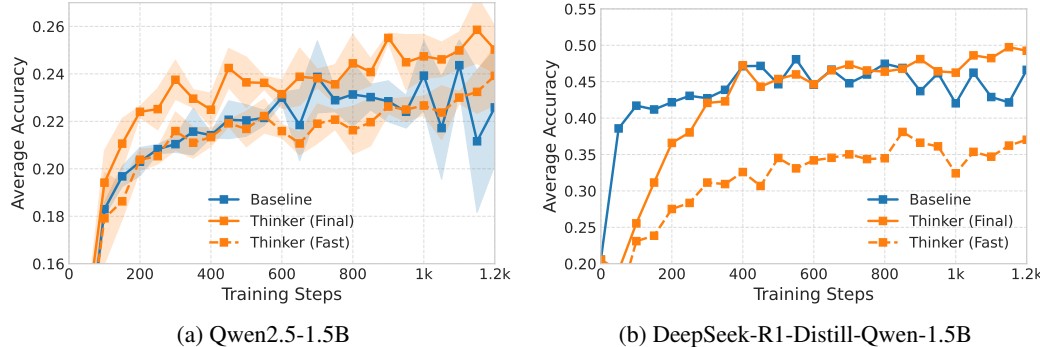

(a) Qwen2.5-1.5B          (b) DeepSeek-R1-Distill-Qwen-1.5B

Figure 5: Evaluation performance across seven common benchmarks. For the Qwen2.5-1.5B model (a), the shaded region represents the standard deviation across three independent seeds.

Table 1: Performance comparison across various mathematical reasoning benchmarks. Average (Avg.) scores are presented. All scores are Pass@1 accuracy (%) averaged over 16 samples. Top score in each benchmark column is **bolded**. Standard errors and more statistical analysis are provided in Appendix B. Results for the Q1.5B Thinker and Baseline are averaged across three independent seeds.

| Method | MATH 500 | AIME 2024 | AIME 2025 | GPQA Diamond | Olympiad bench | AMC 23 | Minerva Math | College Math | Avg. |
|---|---|---|---|---|---|---|---|---|---|
| **Qwen2.5-1.5B (Q1.5B)** | | | | | | | | | |
| Pretrained | 9.05 | 0.00 | 0.00 | 4.55 | 3.09 | 4.06 | 2.30 | 7.40 | 3.81 |
| Baseline | 59.82 | 4.10 | **2.43** | 20.52 | 26.05 | 35.36 | 19.25 | 37.42 | 25.62 |
| Thinker | **64.45** | **6.25** | 2.22 | 19.21 | **28.21** | **39.06** | **20.38** | **38.82** | **27.33** |
| Thinker-Fast | 59.82 | 4.58 | 1.25 | **21.28** | 24.52 | 34.53 | 17.85 | 37.58 | 25.18 |
| ORZ | 58.00 | 3.50 | 1.00 | 16.80 | - | - | - | - | - |
| SimpleRL | 59.00 | 4.20 | - | - | 21.00 | 35.00 | 20.20 | - | - |
| **DeepSeek-R1-Distill-Qwen-1.5B (R1.5B)** | | | | | | | | | |
| Pretrained | 76.21 | 17.50 | 17.92 | 13.76 | 37.46 | 55.94 | 24.82 | 38.85 | 35.31 |
| Baseline | 86.24 | 35.42 | 23.75 | 25.69 | 49.22 | 72.81 | 32.08 | 42.02 | 45.90 |
| Thinker | **88.51** | **38.96** | **26.67** | **37.41** | **55.49** | **83.59** | **34.77** | **42.46** | **50.98** |
| Thinker-Fast | 81.35 | 18.33 | 14.58 | 28.85 | 45.68 | 66.41 | 31.39 | 41.74 | 41.05 |

used to assess intermediate checkpoints. For a comprehensive comparison, the table includes the Fast and Final Accuracy of our Thinker agent, the accuracy of the fine-tuned Baseline, and results for the original Pretrained model. Since evaluation is performed in inference mode, the Thinker agent does not require external verification, making its Final Accuracy directly comparable to the single-attempt baselines. Furthermore, we include reported figures from Open-Reasoner-Zero (ORZ) [3] and SimpleRL [2], both of which also utilize PPO to train agents on standard QA tasks.

From Table 1, we observe that the Thinker agent consistently performs better than the other methods across almost all benchmarks. For the Q1.5B model, the Thinker-Fast agent's performance is already close to that of the baseline, while the full Thinker agent achieves a 6.7% relative performance gain on average compared to this baseline. For the R1.5B model, the Thinker-Fast agent performs slightly worse than the baseline but still significantly outperforms the pretrained model. This result for Thinker-Fast is notable, suggesting a substantial efficiency gain, as it only requires 1000 token budget compared to the 8000 token budget of the pretrained model. The full Thinker-task model with R1.5B surpasses the baseline by an average relative increase of 11.1%. These results collectively suggest the benefits of decomposing the single-turn QA task into the proposed four-step Thinker task. To confirm these benefits extend to larger models, additional experiments on 7B models were conducted, yielding similar results as detailed in Appendix D.

Table 2: Comparison with concurrent works fine-tuning R1.5B models on token efficiency benchmarks. Results from concurrent works are extracted from the respective papers.

| Method | MATH500 | | AIME24 | | AMC23 | |
|---|---|---|---|---|---|---|
| | Acc. (%) | Length | Acc. (%) | Length | Acc. (%) | Length |
| ThinkPrune [19] | 83.2 | 1938 | 27.1 | 5631 | 73.2 | 3039 |
| Concise Reasoning [20] | 81.0 | 1965 | 30.0 | 6752 | 69.4 | 2936 |
| SR-FLOW [21] | 85.3 | - | 36.7 | - | 77.8 | - |
| AdaptThink [22] | 82.0 | 1782 | 31.0 | 6679 | - | - |
| Baseline | 86.2 | 2780 | 35.4 | 5778 | 72.8 | 3938 |
| Thinker | **88.5** | 2501 | **39.0** | 5597 | **83.6** | 3517 |
| Thinker-Fast | 80.9 | 600 | 18.1 | 853 | 66.9 | 751 |

To evaluate token efficiency, we analyze the Thinker agent on three benchmarks used in concurrent work on this topic (Table 2). The results show that our agent achieves superior performance across all three benchmarks with a competitive token count. The Thinker-Fast mode is also effective on easier datasets like MATH500 and AMC23, using substantially fewer tokens. This highlights a key advantage for deployment: the flexibility to select the appropriate mode based on the task's complexity.

## 5.3 Analysis and Case Study

**Reflection and Response Length.** Beyond overall performance, we investigated whether the Thinker task encourages more direct reasoning with less overt self-reflection or self-doubt. We compiled a vocabulary list of terms commonly associated with self-reflection in Deepseek R1's outputs (e.g., "wait," "however," "alternatively") and measured the average occurrence of these *reflection patterns* during training, with results shown in Figure 6a. We observed that the Thinker agent tends to use fewer such reflection patterns in its reasoning trace, suggesting it may be learning to reason more directly.

A detailed breakdown of response length during training is presented in Figure 6b. The trends for the two agents diverge: after stabilizing, the baseline's response length steadily increases, likely due to an increase in learned self-reflection. Conversely, the Thinker agent's total response length gradually decreases. This improved efficiency is not due to a reduction in the length of individual stages, which remain consistent. Rather, it stems directly from the model's rising Fast Thinking accuracy. As the agent's intuition improves, it solves more problems correctly in the Fast Thinking stage, thereby allowing it to bypass the token-intensive Slow Thinking stage and significantly reduce its average token length.

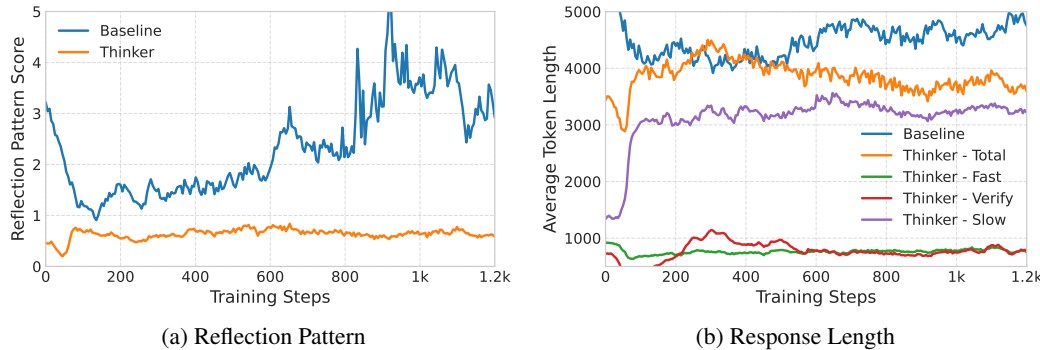

(a) Reflection Pattern                    (b) Response Length

Figure 6: Average reflection pattern count and response length for DeepSeek-R1-Distill-Qwen-1.5B (R1.5B) during training.

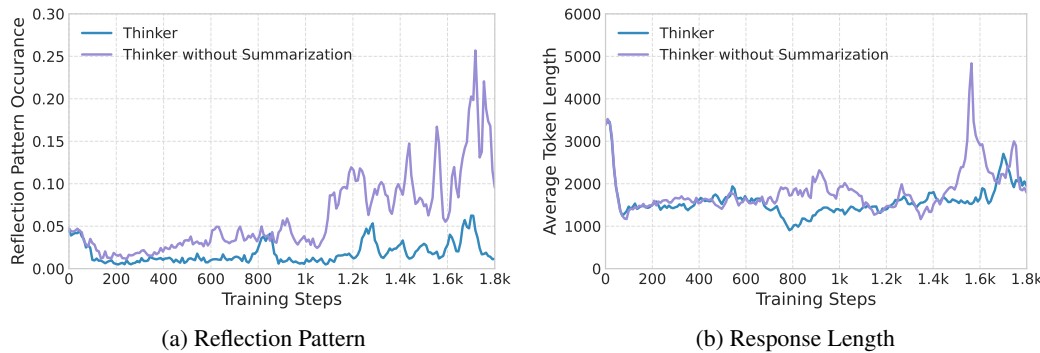

|   |   |
|---|---|
| (a) Reflection Pattern | (b) Response Length |

Figure 7: Average reflection pattern count and response length for Qwen2.5-1.5B (Q1.5B) during training, with and without Summarization stage.

**Role of Summarization.**[2]   Finally, we investigated the importance of the Summarization step through an ablation analysis. We trained a Q1.5B model using a modified Thinker task where the Summarization step was removed. The average reflection pattern count and response length for the Thinker task, both with and without the Summarization step, are presented in Figure 7. Without the Summarization step, the model exhibited a more frequent increase in reflection patterns, accompanied by highly fluctuating response lengths. This indicates that the Summarization step may contribute to stabilizing the learning process, potentially by discouraging degeneration into excessive reflection, though we cannot rule out that the observed divergence is due to stochastic variation.

Removing the Summarization step negatively impacted Fast Thinking: average Fast Accuracy dropped from 26.75% to 24.84%. This suggests summarization, by potentially distilling concise reasoning, enhances intuition for Fast Thinking. Interestingly, final accuracy remained comparable (27.85% with vs. 27.84% without summarization), indicating that the Slow Thinking step could often compensate for the reduced fast accuracy. Detailed ablation results can be found in Appendix B.

**Case Study.** We conducted a case study to examine how the agent's reasoning adapts across the Thinker task stages. Observations from representative outputs (see Appendix C for full examples) highlight the importance of the Fast Thinking mode. When an initial Fast Thinking output is incorrect, the agent first engages in a Verification stage to assess its own answer with clear reference to the steps given in the Fast Thinking mode. If errors are identified, the subsequent Slow Thinking stage involves a detailed refinement. During Slow Thinking, the agent *explicitly* scrutinizes the flawed reasoning from the Fast Thinking attempt and insights from its own Verification, endeavoring to generate a new, correct solution. Furthermore, after a successful correction, the agent learns to distill the long reasoning into concise explanations during the last Summarization stage.

## 6   Future Work and Conclusion

In this paper, we introduced the Thinker task, a novel approach that decomposes the standard single-turn question-answering process into a structured four-stage task. This decomposition aims to provide more explicit reward signals for training distinct agent capabilities: intuition via Fast Thinking, evaluation via Verification, refinement via Slow Thinking, and integration via Summarization.

Beyond the application to question-answering, this work underscores the potential of environment augmentation in RL. In RL, while significant attention is devoted to algorithm development, the environment itself is typically treated as a given problem specification. However, as demonstrated by this paper and prior research such as the Thinker MDP, there is considerable untapped potential in designing environments that offer richer inputs, more structured interactions, or more nuanced reward signals. Future research could explore alternative methods for enriching RL environments, perhaps by developing more dynamic tasks that adapt to an agent's learning state or that explicitly target the development of a wider array of cognitive capabilities. Such advancements in environment design could unlock new levels of performance and emergent abilities in RL agents.

---

[2]This ablation study used a 6000-token Verification step. The length was later reduced to 2000 tokens for the main experiments, as this did not significantly affect performance.

## Acknowledgment

Jie Fu is supported by Shanghai Artificial Intelligence Laboratory.

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

# A  Experimental Details

This section describes the details of the experiment in Section 5. Our implementation, adapted from the Open-Reasoner-Zero codebase [3], is publicly available at `https://github.com/stephen-chung-mh/thinker-task`, which also includes the trained models.

## A.1  Hyperparameters

We use the same hyperparameters as those provided in Open-Reasoner-Zero [3], except that we reduce the number of samples per prompt from 64 to 32 to save computational resources. One training step proceeds as follows: we first randomly sample 128 prompts (rollout batch size) from the training dataset and generate 32 samples per prompt, totaling $128 \times 32 = 4{,}096$ samples. We then divide the generated samples into 1 (12) training batch for the actor (critic), where each training batch is used for one optimizer update.

We tune the coefficient $c$ in $R_{\text{summary}}$ by searching over $\{1\text{e-}4, 1\text{e-}3, 1\text{e-}2\}$. Other Thinker-task-specific hyperparameters are selected using heuristics. We use a lower sampling temperature during summarization, as we observe that higher temperatures tend to produce less concise and consistent summaries.

For the baseline model, we use similar hyperparameters, except with a generation length of 8,000 tokens. We found that 8,000 tokens yield optimal baseline performance on R1.5B.

Table 3: Hyperparameters used in experiments.

| Parameter | Value |
| --- | --- |
| **PPO** | |
| Rollout Batch Size | 128 |
| Number of Samples Per Prompt | 32 |
| Number of Epochs | 1 |
| Actor Learning Rate | 1e-6 |
| Number of Actor Update Steps | 1 |
| Critic Learning Rate | 5e-6 |
| Number of Critic Update Steps | 12 |
| Discount Rate $\gamma$ | 1 |
| GAE Lambda $\lambda$ | 1 |
| Clip Ratio $\epsilon$ | 0.2 |
| KL Loss | None |
| Sampling Temperature | 1 |
| Sampling Temperature in Summarization | 0.6 |
| | |
| **Generation Length** | |
| Fast Thinking | 1,000 |
| Verification | 2,000 |
| Slow Thinking | 6,000 |
| Summarization | 1,000 |
| | |
| **Reward-specific** | |
| Coefficient $c$ in $R_{\text{summary}}$ | 1e-3 |
| Minimum Length for Summarization | 300 |

## A.2  Prompt Templates

The prompt templates used in the four stages of the Thinker task are illustrated in Box A.1. Note that not all prompts are necessarily used. For example, in training mode, if the agent's fast answer is correct, the Slow Thinking and Summarization prompt will be skipped. Please refer to the main text for the termination conditions.

> **Box A.1: Prompt Templates for Thinker Task**
>
> **1. Fast Thinking**
>
> **User:** Answer the below question with concise steps and output the final answer within \boxed{}. Limit your response below 1000 words.
> This is the problem: {question}
> **Assistant:** *<Agent Response>*
>
> **2. Verification**
>
> **User:** Is your answer above correct? Please verify each step and the answer carefully. Output \boxed{Yes} if your answer is correct, or \boxed{No} if your answer is incorrect.
> **Assistant:** *<Agent Response>*
>
> **3. Slow Thinking**
>
> **User:** Your initial answer is incorrect. Now, think about the possible errors and consider alternative solutions. The reasoning process should be enclosed within <think>...</think>.
> This is the problem: {question}
> Let's think step by step and output the final answer within \boxed{}.
> **Assistant:** <think> *<Agent Response>*
>
> **4. Summarization**
>
> **User:** Your final answer is correct. Now summarize the steps leading to your final answer concisely and precisely, excluding internal reasoning. Limit your response between 300 and 1000 words.
> This is the problem: {question}
> **Assistant:** *<Agent Response>*

## A.3 Computational Resources

Each training run for both the Thinker task and the baseline required approximately 7 days on two compute nodes, each equipped with 8 A100 GPUs. We use the Deepspeed [27], vLLM [28], and Ray [29] library for distributed training.

# B  Result Details

This section describes additional experimental results that were omitted from the main text due to length constraints.

## B.1  Evaluation Results

Figure 8 and Figure 9 show the breakdown of Figure 5 from the main text, corresponding to the evaluation results of fine-tuning Q1.5B and R1.5B on the QA task or the Thinker task during training.

The detailed evaluation results of the ablated run in which the Summarization step is removed can be found in Table 4, labeled as `SkipSum`.

Table 5 and Table 6 present the standard error corresponding to the results reported in Table 1. The standard errors here are computed using Equation (1) from [30].

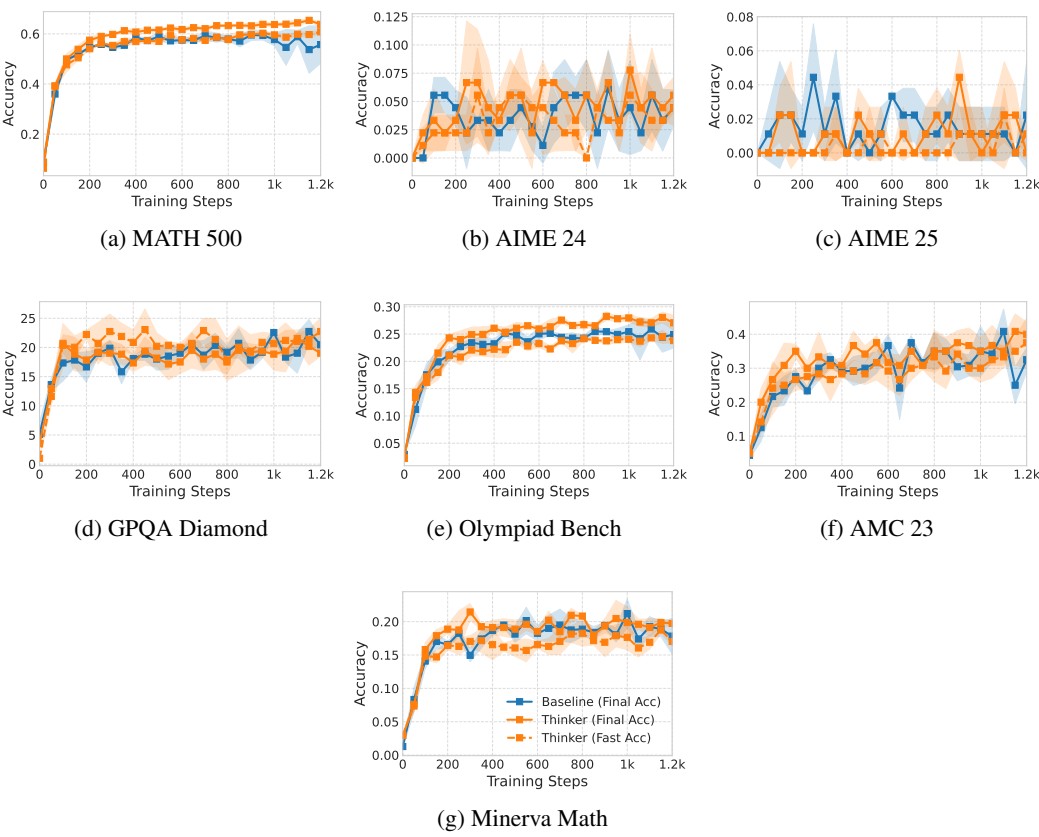

Figure 8: Detailed evaluation results of Q1.5B fine-tuned using QA task or Thinker task on individual mathematical reasoning benchmarks. The shaded region represents the standard deviation across three independent seeds

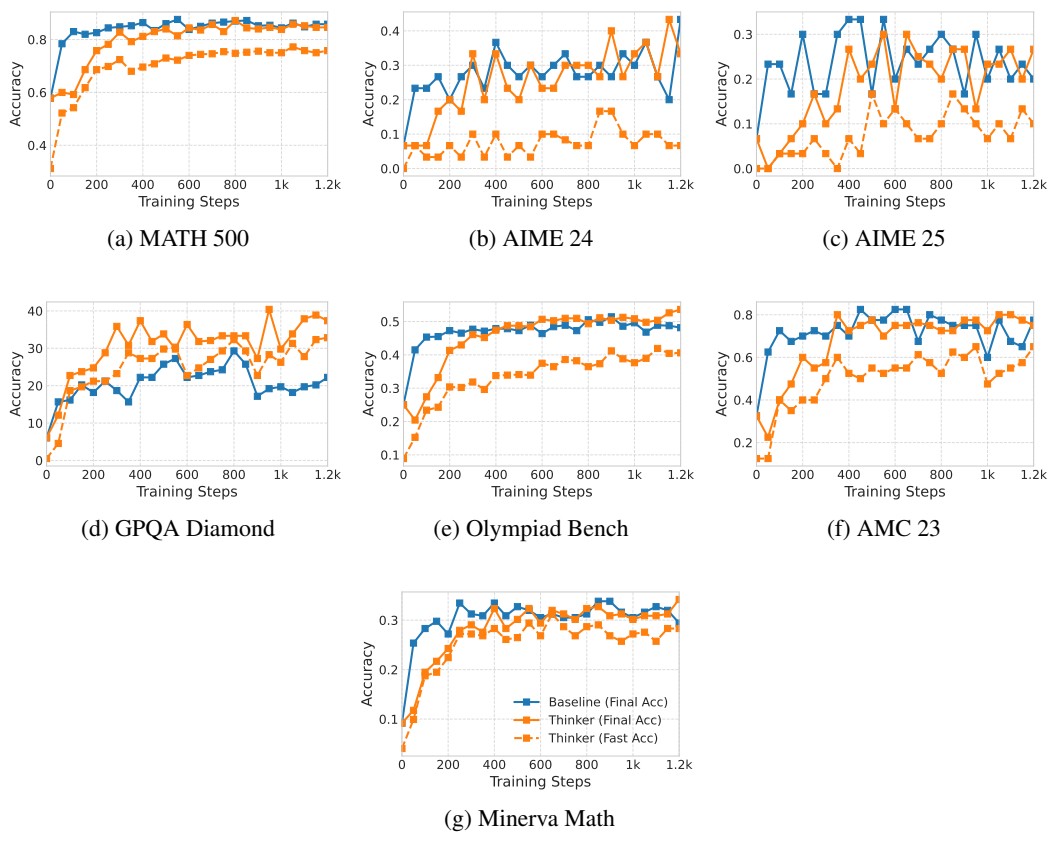

(a) MATH 500      (b) AIME 24      (c) AIME 25

(d) GPQA Diamond      (e) Olympiad Bench      (f) AMC 23

(g) Minerva Math

Figure 9: Detailed evaluation results of R1.5B fine-tuned using QA task or Thinker task on individual mathematical reasoning benchmarks.

Table 4: Mathematical reasoning performance of the Thinker agent trained without the Summarization stage. Average (Avg.) scores are presented. All scores are Pass@1 accuracy (%) averaged over 16 samples. All runs used a 6000-token Verification step.

| Method | MATH 500 | AIME 2024 | AIME 2025 | GPQA Diamond | Olympiad bench | AMC 23 | Minerva Math | College Math | Avg. |
|---|---|---|---|---|---|---|---|---|---|
| **Qwen2.5-1.5B (Q1.5B)** | | | | | | | | | |
| Thinker | **64.25** | 6.25 | 2.50 | 23.74 | 28.11 | **40.62** | 19.03 | 38.33 | 27.85 |
| Thinker-Fast | 61.60 | 6.25 | 2.50 | **26.39** | 24.78 | 35.94 | 18.66 | 37.85 | 26.75 |
| SkipSum | 64.30 | **9.17** | **4.17** | 18.62 | **29.11** | 37.50 | **20.82** | **39.42** | **27.89** |
| SkipSum-Fast | 60.30 | 5.00 | 1.25 | 20.27 | 24.17 | 30.00 | 19.85 | 38.24 | 24.88 |

Table 5: Standard error analysis for the Q1.5BB model. All scores are in %. The values are presented as score (standard error).

| Benchmark | # Qs | Thinker | | | Thinker-Fast | | | Baseline | | |
|---|---|---|---|---|---|---|---|---|---|---|
| | | Seed 1 | Seed 2 | Seed 3 | Seed 1 | Seed 2 | Seed 3 | Seed 1 | Seed 2 | Seed 3 |
| MATH 500 | 500 | 63.25 | 64.72 | **65.38** | 59.42 | 59.67 | 60.38 | 57.98 | 60.72 | 60.75 |
| | | (1.83) | (1.79) | **(1.80)** | (1.83) | (1.85) | (1.84) | (1.79) | (1.80) | (1.79) |
| AIME 2024 | 30 | 5.62 | 5.83 | **7.29** | 5.00 | 4.17 | 4.58 | 3.33 | 4.38 | 4.58 |
| | | (2.51) | (3.20) | **(3.34)** | (2.83) | (2.69) | (3.16) | (1.58) | (2.61) | (2.49) |
| AIME 2025 | 30 | **3.54** | 1.88 | 1.25 | 1.46 | 1.04 | 1.25 | 3.33 | 1.46 | 2.50 |
| | | **(1.84)** | (1.13) | (0.70) | (1.26) | (0.85) | (0.63) | (1.79) | (1.07) | (1.36) |
| GPQA Diamond | 198 | 19.00 | 17.99 | 20.64 | **23.11** | 19.85 | 20.86 | 21.46 | 19.35 | 20.74 |
| | | (1.69) | (1.67) | (1.78) | **(1.77)** | (1.84) | (1.81) | (1.80) | (1.61) | (1.74) |
| Olympiad Bench | 675 | 28.02 | 27.94 | **28.67** | 24.82 | 24.13 | 24.60 | 24.54 | 26.46 | 27.16 |
| | | (1.44) | (1.45) | **(1.46)** | (1.35) | (1.37) | (1.35) | (1.32) | (1.39) | (1.38) |
| AMC23 | 40 | 38.91 | **41.41** | 36.88 | 36.09 | 34.84 | 32.66 | 34.38 | 36.41 | 35.31 |
| | | (6.11) | **(6.38)** | (6.18) | (6.43) | (5.74) | (5.78) | (5.62) | (5.67) | (5.46) |
| Minerva Math | 272 | 19.90 | **21.83** | 19.42 | 16.98 | 20.06 | 16.50 | 17.78 | 19.88 | 20.08 |
| | | (1.92) | **(2.02)** | (1.90) | (1.75) | (1.90) | (1.74) | (1.85) | (1.88) | (1.94) |
| College Math | 2818 | 38.73 | **39.15** | 38.58 | 37.49 | 37.84 | 37.39 | 35.85 | 38.26 | 38.15 |
| | | (0.85) | **(0.86)** | (0.85) | (0.85) | (0.84) | (0.84) | (0.81) | (0.84) | (0.85) |

Table 6: Standard error analysis of R1.5B models on mathematical benchmarks. All scores are in %. The values are presented as score (standard error).

| Benchmark | # Questions | Thinker | Thinker-Fast | Baseline |
|---|---|---|---|---|
| MATH 500 | 500 | **88.51 (1.22)** | 81.35 (1.50) | 86.24 (1.25) |
| AIME 2024 | 30 | **38.96 (7.21)** | 18.33 (5.23) | 35.42 (7.09) |
| AIME 2025 | 30 | **26.67 (7.52)** | 14.58 (5.37) | 23.75 (6.74) |
| GPQA Diamond | 198 | **37.41 (2.24)** | 28.85 (2.27) | 25.69 (2.00) |
| Olympiad Bench | 675 | **55.49 (1.67)** | 45.68 (1.66) | 49.22 (1.66) |
| AMC23 | 40 | **83.59 (4.70)** | 66.41 (5.94) | 72.81 (5.18) |
| Minerva Math | 272 | **34.77 (2.57)** | 31.39 (2.43) | 32.08 (2.47) |
| College Math | 2781 | **42.46 (0.90)** | 41.74 (0.89) | 42.02 (0.89) |

## B.2 Ablation Study on Fast Thinking Mode

To understand the importance of the Fast Thinking mode in the overall Thinker task, we experiment by using a less-trained agent to generate the Fast Thinking response, while still using the fully trained agent to generate responses for the remaining stages.[3] This allows us to measure the impact of Fast Thinking quality on overall performance.

Specifically, we use four earlier R1.5B Thinker-agent checkpoints (Step 0, which is the pretrained model; Step 200; Step 400; and Step 600) to generate the Fast Thinking response, and use the fully trained R1.5B Thinker-agent for the remaining stages. We evaluate final accuracy across the eight benchmarks, as in the main evaluation. The results are shown in Figure 10.

We observe a general positive correlation between the Fast Thinking accuracy of a checkpoint and the final accuracy, suggesting that the Fast Thinking response has a substantial influence on subsequent stages. For instance, when we use the pretrained model (Step 0) to generate the Fast Thinking response, final accuracy drops significantly from 49.8% to 36.3%. However, we also observe that this sensitivity diminishes as Fast Thinking performance improves. For example, using the Step 200 model, which has a moderate Fast Thinking accuracy of 28.9%, leads to a final performance of 49.18%—a minor drop from 49.8%.

We conjecture that this is due to the robustness of the Slow Thinking mode: since it is trained specifically to handle incorrect Fast Thinking answers, it can often recover from slightly flawed initial intuition. However, if the Fast Thinking intuition is very poor (as in the pretrained model), the subsequent stages may fail to recover due to the lack of a meaningful starting point. A qualitative analysis of how the Fast Thinking stage interacts with subsequent stages can be found in the case study in Appendix C, which shows that the trained agent is able to correct flawed heuristics from the Fast Thinking mode during the Verification and Slow Thinking mode.

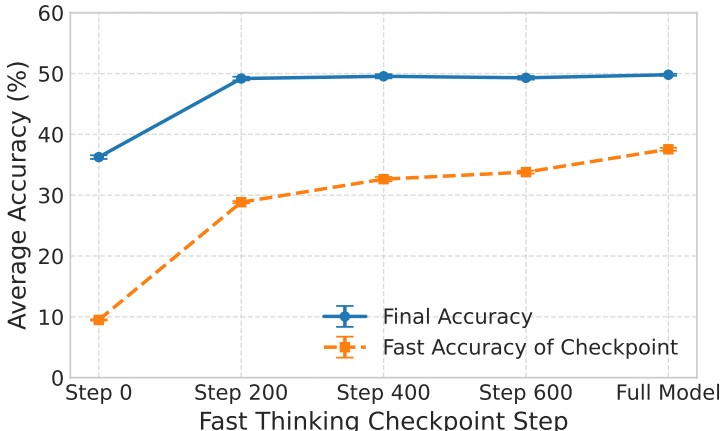

Figure 10: Final accuracy on the evaluation benchmarks of the Thinker agent (R1.5B) when its Fast Thinking stage is generated by model checkpoints at previous training steps. The original Fast Thinking accuracy of these respective checkpoints is also shown. All scores are Pass@1 accuracy (%) averaged over 16 samples. Error bars represent standard error, which are typically minor in this data.

---

[3]The section here used a 6000-token Verification step. The length was later reduced to 2000 tokens for the main experiments, as this did not significantly affect performance.

# C   Case Study

In this section, we present sampled responses from the fine-tuned R1.5B agent on the Thinker task, aiming to understand the behavior learned by the agent. Only responses with an incorrect fast answer are selected, so that the interaction between Fast Thinking and Slow Thinking can be observed.

## C.1   Case Study I: Identifying Flaws in Fast Thinking (Box C.1, Box C.2)

This example demonstrates the Thinker task's ability to guide the agent from an uncertain, flawed initial attempt to a correct, verified solution by structuring its reasoning process.

**Fast Thinking.** The agent adopts a quick, System-1-like heuristic despite expressed uncertainty ("the side length of the larger hexagon is $a + 2 \cdot 2 = a + 4$? Not sure.") and proceeds with this flawed assumption ($S = a + 4$). This leads to an incorrect perimeter, $6\sqrt{3} - 12$, which is physically implausible as it results in a negative value.

**Verification.** The agent directly confronts this flawed assumption. It explicitly questions the initial logic ("The side length of this larger hexagon might be $s + 4$? Or is it $s + 2$? Wait, reconsider. <...> the relationship isn't straightforward."). This critical re-evaluation leads to the correct geometric insight, yielding the relationship $S = s + \frac{\sqrt{3}}{4}$. Based on this corrected understanding, the agent identifies the error in its initial reasoning and conclusion, stating, "The previously given answer was $6\sqrt{3} - 12$. But that would not match. <...> Thus our initial approach is wrong."

**Slow Thinking.** The agent then leverages the insight from Verification. It explicitly focuses on the "difference in the apothems" to re-derive $S = s + \frac{\sqrt{3}}{4}$. This demonstrates a clear adoption of the successful reasoning trace from Verification. The agent then systematically solves for the side length $s$ and calculates the correct perimeter, $18 - 4\sqrt{3}$. Notably, it independently performs a numerical check, showcasing a deeper level of deliberation and confidence in its refined answer.

**Summarization.** The provided summary effectively distills the core mathematical steps for solving the problem into a clear, concise, and logically consistent sequence. It accurately establishes the relationship between the side lengths of the inner and outer hexagons, correctly formulates the equation for the path's area, and finds the pool's perimeter efficiently. Interestingly, it also employs a thinking block that reflects certain self-correction patterns observed in earlier steps.

This case highlights the agent's capacity for targeted error identification and conceptual correction. The progression shows a clear refinement of reasoning, moving from the System-1-like heuristic in Fast Thinking to a more rigorous, System-2-like approach in Verification and Slow Thinking. The explicit references between stages—Verification critiquing Fast Thinking's "initial approach" and Slow Thinking building directly on Verification's apothem insight—underscore the efficacy of the structured task in fostering coherent, self-correcting thought processes.

## C.2   Case Study II: Propagation of Error from Verification to Slow Thinking (Box C.3)

This example provides a counterpoint to the previous successful error-correction cases. It demonstrates a scenario in which the agent arrives at an incorrect final answer due to error propagation and insufficient depth in later-stage reasoning.

**Fast Thinking.** The agent, faced with a complex product of fractional parts, makes a guess. After calculating the first few terms and noting the initial product starts with $2 \cdot \frac{1}{2} = 1$, it states: "Since initial term is 2 and product involves fractions potentially leading to 1/2." This leads to the incorrect answer D ($\frac{1}{2}$), driven more by the available options and a superficial pattern than a rigorous derivation.

**Verification.** This stage correctly identifies Fast Thinking's answer as likely wrong. However, it introduces its own flawed heuristic: "Since each term $\frac{a_k}{a_{k-1}}$ is less than 1, and multiple such terms multiplied, so perhaps the total product tends to zero." While it correctly prompts a re-evaluation by concluding the original answer was incorrect, its reasoning toward an alternative answer (0) is based on a wrong heuristics without considering the specific structure of this sequence (which actually involves a telescoping product that simplifies to $\frac{a_1}{a_{98}}$, leading to a final answer of 1).

**Slow Thinking.** Instead of uncovering the correct algebraic simplification, Slow Thinking explicitly adopts and reinforces the flawed heuristic from Verification. After refining some calculations, it

reasons: "Alternatively, since the product seems to be approaching zero due to the product of many terms each less than 1. But given the options, perhaps it's 0." This echoes Verification's logic and leads to the incorrect final answer B (0).

This case demonstrates how flawed reasoning can persist within the Thinker task. While Verification correctly identified Fast Thinking's initial guess as wrong, it unfortunately introduced its own plausible but incorrect heuristic. Subsequently, Slow Thinking, anchored on this flawed premise inherited from Verification, performing more calculations around it rather than challenging its validity. This highlights the possibility of an anchoring bias on an incorrect heuristic, and ultimately, superficial deliberation in the later stages that fails to correct the core flawed assumption.

---

**Box C.1: Example Response from Thinker Agent: Hexagon Path Problem**

**Problem:** A path 2 m wide and with an area of 36 m$^2$ has been built around a swimming pool in the shape of a regular hexagon. What is the perimeter of the pool?

**User:** <Fast Thinking Prompt: Answer concisely...>
**Assistant:**
<Initial Reasoning Trace showing uncertainty about side length>
3. <...> **the side length of the larger hexagon is** $a + 2 \cdot 2 = a + 4$**? Not sure.**
<Calculations based on the assumption $S = a + 4$>
13. So $8a = 8\sqrt{3} - 16 \Rightarrow a = \sqrt{3} - 2$.
14. Perimeter of the hexagon is $6a = 6(\sqrt{3} - 2) = 6\sqrt{3} - 12$.

$\boxed{6\sqrt{3} - 12}$

**User:** <Verification Prompt: Is your answer above correct?>
**Assistant:**
<Reasoning Trace in Verification, questioning the $S = s + 4$ assumption ($a$ relabeled as $s$)>
**The side length of this larger hexagon might be** $s + 4$**? Or is it** $s + 2$**?**
Wait, reconsider. <...> the relationship isn't straightforward.
<Derivation of the correct relationship $S = s + \frac{4}{\sqrt{3}}$ based on apothems>
<Calculations using the corrected $S$, leading to Perimeter $= 18 - 4\sqrt{3}$>
The previously given answer was $6\sqrt{3} - 12$.
But that would not match. So our approach seems incorrect.
<...> **Thus our initial approach is wrong.** Therefore, the original answer is incorrect.
<...> $\boxed{No}$

**User:** <Slow Thinking Prompt: Your initial answer is incorrect. Now, think about errors...>
**Assistant:** `<think>`
<revisiting the core geometric error from Fast Thinking>
When you add a path around the hexagon, **the increase in side length can be found by considering the distance from the center**. <...>
The path is 2m wide, which I think refers to the distance from the pool to the path, that would mean the difference in the apothems is 2m. So:
$\frac{S\sqrt{3}}{2} - \frac{s\sqrt{3}}{2} = 2$
<Re-derivation of $S = s + \frac{4}{\sqrt{3}}$, consistent with Verification's insight>
$S = s + \frac{4}{\sqrt{3}}$
<Systematic algebraic solution using the corrected $S$, leading to $s = 3 - \frac{2\sqrt{3}}{3}$>
<Calculation of perimeter $= 18 - 4\sqrt{3}$, followed by numerical verification>
Hence, the perimeter is $6s \approx 18 - 4\sqrt{3}$ meters. <...> Yes, that seems consistent.

Final Answer. The perimeter of the pool is $\boxed{18 - 4\sqrt{3}}$ meters.
`</think>`
<Narrative solution in Slow Thinking, presenting the final corrected steps>

Box C.2: Example Summary from Thinker Agent: Hexagon Path Problem

**Problem:** A path 2 m wide and with an area of 36 m$^2$ has been built around a swimming pool in the shape of a regular hexagon. What is the perimeter of the pool?

**User:** <Summarization Prompt: Summarize concisely...>
**Assistant:**
<think>
1. Let s be the side length of the original hexagonal pool. The path surrounds the pool and forms a larger hexagon. The width of the path is 2m, so the side length of the larger hexagon is $s + 2 \cdot (2/\sqrt{3}) = s + 4/\sqrt{3}$. Wait, actually, the relationship is more complex.
2. The area of the path is the difference between the area of the larger hexagon and the smaller one. The area of a regular hexagon is $(3\sqrt{3}/2) \cdot$ (side length)$^2$.
3. Let s be the side length of the pool, then the side length of the larger hexagon is $s + 2 \cdot (2/\sqrt{3}) = s + 4/\sqrt{3}$. Wait, no, the correct relationship is that the side length increases by $2 \cdot (2/\sqrt{3}) = 4/\sqrt{3}$.
4. The area of the path is $36 = (3\sqrt{3}/2) \cdot [(s + 4/\sqrt{3})^2 - s^2]$.
5. Simplifying the equation, we find that $s = 3 - (2\sqrt{3})/3$. The perimeter is $6 \cdot s = 18 - 4\sqrt{3}$.

Thus, the perimeter of the pool is $\boxed{18 - 4\sqrt{3}}$ meters.
</think>

Let $s$ be the side length of the original hexagonal pool. The path width is 2m. The side length of the larger hexagon (pool + path) is $S = s + \frac{4}{\sqrt{3}}$. The area of the path is 36 m$^2$, which is the difference between the areas of the larger and smaller hexagons.

The area of a regular hexagon is $\frac{3\sqrt{3}}{2} \times$ (side length)$^2$. Therefore:

$$\frac{3\sqrt{3}}{2}\left(\left(s + \frac{4}{\sqrt{3}}\right)^2 - s^2\right) = 36$$

Simplifying the term inside the brackets:

$$\left(s + \frac{4}{\sqrt{3}}\right)^2 - s^2 = s^2 + \frac{8s}{\sqrt{3}} + \frac{16}{3} - s^2 = \frac{8s}{\sqrt{3}} + \frac{16}{3}$$

Substituting this back into the area equation:

$$\frac{3\sqrt{3}}{2}\left(\frac{8s}{\sqrt{3}} + \frac{16}{3}\right) = 36$$

Multiplying through by $\frac{3\sqrt{3}}{2}$:

$$\frac{3\sqrt{3}}{2} \cdot \frac{8s}{\sqrt{3}} + \frac{3\sqrt{3}}{2} \cdot \frac{16}{3} = 36$$

$$12s + 8\sqrt{3} = 36$$

Solving for $s$:

$$12s = 36 - 8\sqrt{3}$$

$$s = \frac{36 - 8\sqrt{3}}{12} = 3 - \frac{2\sqrt{3}}{3}$$

The perimeter of the pool is $6s$:

$$6\left(3 - \frac{2\sqrt{3}}{3}\right) = 18 - 4\sqrt{3}$$

Thus, the perimeter of the pool is $\boxed{18 - 4\sqrt{3}}$ meters.

> **Box C.3: Example Response from Thinker Agent: Sequence Product Problem**
>
> **Problem:** Define the sequence $\{a_n\} : a_1 = 1, a_2 = 2, a_{n+2} = a_n + a_{n+1}, n \in \mathbf{N}_+$, then find $\left[\frac{a_2}{a_1}\right] \cdot \left\{\frac{a_3}{a_2}\right\} \cdot \ldots \cdot \left\{\frac{a_{99}}{a_{98}}\right\} \cdot \left[\frac{a_{98}}{a_2}\right]$. Options: A. 2, B. 0, C. 1, D. $\frac{1}{2}$
>
> **User:** <Fast Thinking Prompt: Answer concisely...>
> **Assistant:**
> <Initial computation of sequence terms and first few product terms: $[a_2/a_1] = 2$, $\{a_3/a_2\} = 1/2$>
> 7. Notice that the product of fractional parts potentially telescopes or simplifies. However, it's complex.
> 8. Given the options A to D, and considering the pattern, likely answer is B.0 or C.1, but need to think carefully. **Since initial term is 2 and product involves fractions potentially leading to 1/2.**
> $\boxed{D}$
> **User:** <Verification Prompt: Is your answer above correct?>
> **Assistant:**
> <Re-evaluation of sequence and product structure, calculating $[a_2/a_1] = 2$ and $[a_{98}/a_2] \approx 17$>
> Wait, count the number of fractional parts: <...> 96 terms.
> <...> **Since each term $\{a_k/a_{k-1}\}$ is less than 1, and multiple such terms multiplied, so perhaps the total product tends to zero.**
> Thus, the product is very small, so the overall expression would be close to zero. <...>
> Therefore, the original answer was incorrect, so the correct answer should be $\boxed{No}$, as the initial answer was incorrect.
> **<...> now I think the final answer might be B.0.**
>
> **User:** <Slow Thinking Prompt: Your initial answer is incorrect. Now, think about errors...>
> **Assistant:** <think>
> <Recalculates sequence terms, $[a_2/a_1] = 2$. Identifies $a_n = F_{n+1}$ and that $F_{99}$ is even, refining $[a_{98}/a_2]$ to $F_{99}/2$>
> Given that each ratio $a_k/a_{k-1}$ approaches the golden ratio $\phi \approx 1.618$.
> Thus, each $\{a_k/a_{k-1}\}$ approaches $\phi - 1 = \psi \approx 0.618$.
> Thus, if each term $\{a_k/a_{k-1}\}$ is about $\psi \approx 0.618$.
> Thus, product of 96 terms [error in count, should be 97]: $\psi^{96}$.
> Therefore, product=$2 \cdot (\psi)^{96} \cdot \text{floor}(F_{99}/2)$.
> **<...> Alternatively, since the product seems to be approaching zero due to the product of many terms each less than 1. But given the options, perhaps it's 0.**
> Therefore, the answer is B.0.
> $\boxed{B}$
> </think>
> <Narrative solution, reinforcing the (incorrect) conclusion that the product is 0>

# D  Additional Experiments on Larger Models

To assess the scalability of the Thinker task, we fine-tuned DeepSeek-R1-Distill-Qwen-7B (R7B) [1], a larger version of the R1.5B model that has also undergone distillation using reasoning data from DeepSeek-R1. The same set of hyperparameters and training data from the main experiments were utilized, with the exception of an increased token budget for the verification stage—from 2,000 to 6,000 tokens—to accommodate the more verbose reasoning of larger models. The baseline for this experiment corresponds to fine-tuning the R7B model on the standard QA task.

The average evaluation accuracy across the seven benchmarks (excluding CollegeMath) during training is shown in Figure 11. The detailed performance of the best checkpoints from both runs can be found in Table 7. Similar to the results observed in smaller models, we found that agents trained under the Thinker task consistently perform better than those trained on the baseline QA task

across all benchmarks. The overall performance improved from 54.41% to 59.09%, representing a relative improvement of 8.6%. This suggests that larger models also benefit from the Thinker task. Additionally, we note that Thinker-Fast performance improves from 41.05% in R1.5B to 47.19% in R7B, demonstrating that the Fast Thinking mode scales well with model size.

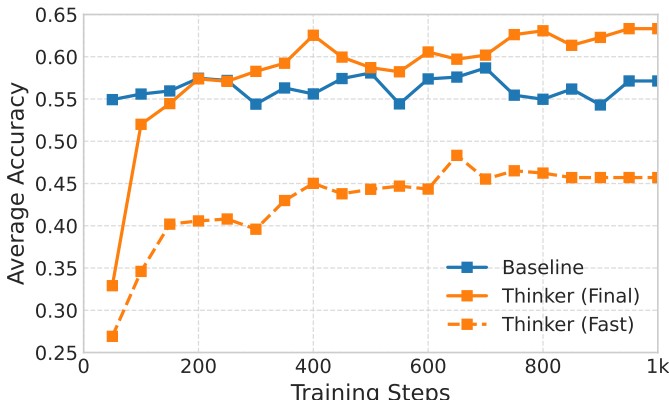

Figure 11: Evaluation performance of R7B averaged across seven common benchmarks.

Table 7: Performance comparison across various mathematical reasoning benchmarks. Average (Avg.) scores are presented. All scores are Pass@1 accuracy (%) averaged over 16 samples. Top score in each benchmark column is **bolded**. Standard errors are provided in Table 8.

| Method | MATH 500 | AIME 2024 | AIME 2025 | GPQA Diamond | Olympiad bench | AMC 23 | Minerva Math | College Math | Avg. |
|---|---|---|---|---|---|---|---|---|---|
| **DeepSeek-R1-Distill-Qwen-7B (R7B)** | | | | | | | | | |
| Pretrained | 84.05 | 37.50 | 28.54 | 17.58 | 37.92 | 36.41 | 34.49 | 40.72 | 39.65 |
| Baseline | 91.03 | 47.50 | 34.58 | 34.63 | 56.76 | 87.81 | 40.23 | 42.71 | 54.41 |
| Thinker | **93.04** | **56.25** | **41.46** | **41.51** | **62.12** | **91.09** | **44.39** | **42.84** | **59.09** |
| Thinker-Fast | 86.47 | 26.46 | 21.88 | 34.12 | 51.77 | 71.56 | 43.08 | 42.14 | 47.19 |

Table 8: Standard error analysis of R7B models on mathematical benchmarks. All scores are in %. The values are presented as score (standard error).

| Benchmark | # Questions | Thinker | Thinker-Fast | Baseline |
|---|---|---|---|---|
| MATH 500 | 500 | **93.04 (0.97)** | 86.47 (1.30) | 91.03 (1.06) |
| AIME 2024 | 30 | **56.25 (7.60)** | 26.46 (6.50) | 47.50 (7.20) |
| AIME 2025 | 30 | **41.46 (7.86)** | 21.88 (6.83) | 34.58 (7.14) |
| GPQA Diamond | 198 | **41.51 (2.31)** | 34.12 (2.54) | 34.63 (2.43) |
| Olympiad Bench | 675 | **62.12 (1.64)** | 51.77 (1.69) | 56.76 (1.63) |
| AMC23 | 40 | **91.09 (3.67)** | 71.56 (6.00) | 87.81 (3.96) |
| Minerva Math | 272 | **44.39 (2.78)** | 43.08 (2.73) | 40.23 (2.71) |
| College Math | 2818 | **42.84 (0.89)** | 42.14 (0.89) | 42.71 (0.88) |

# E  Societal Impacts

This research contributes to enhancing the reasoning capabilities of LLMs, which could positively impact areas like scientific advancement and education. By promoting more structured reasoning through the Thinker task, we aim for AI systems that are not only more performant but also potentially more understandable. However, as LLMs become more powerful, it remains essential to address potential risks, including misuse and unintended societal consequences, through continued research into AI safety, ethics, and governance.

