# OpenReview forum: "Thinker: Learning to Think Fast and Slow"
_NeurIPS.cc/2025/Conference — NeurIPS 2025 poster_

### Official Review · Reviewer_5wyw · 2025-07-01

**Clarity:** 3
**Significance:** 2
**Originality:** 2
**Rating:** 4
**Confidence:** 3

**Summary:**

This paper focuses on improving the reasoning capabilities of large language models (LLMs). One representative model is DeepSeek R1, which performs well but relies heavily on backtracking and verification, often resulting in long and redundant responses. To enhance efficiency, the paper proposes a dual-process framework inspired by cognitive psychology. The method involves four steps: the agent first generates an initial answer rapidly using a small token budget, then evaluates it with a larger token budget. If the answer is deemed incorrect, the agent produces a revised response using more resources. During training, a summarization step is introduced to strengthen the model’s fast-thinking ability. Empirically, the method is evaluated on several mathematical reasoning benchmarks, showing improved accuracy over baseline approaches.

**Questions:**

- On GPQA, why does *Thinker-Fast* outperform the full *Thinker* model? What insights can be drawn from this result?

**Ethical Concerns:**

["NO or VERY MINOR ethics concerns only"]

**Final Justification:**

The author has addressed my questions in the rebuttal. So I think a weak accept is appropriate for this paper.

**Limitations:**

yes

**Paper Formatting Concerns:**

no issue

**Quality:**

3

**Strengths And Weaknesses:**

**Strengths:**

- Improving the reasoning abilities of LLMs through reinforcement learning is a highly active and relevant research direction, especially following the success of DeepSeek R1. This paper builds upon that line of work by decoupling reasoning into two complementary systems: a *fast thinking* component for intuitive response generation and a *slow thinking* component for verification and deliberation. The motivation is grounded in cognitive psychology, and the empirical results demonstrate the effectiveness of the proposed dual-process framework.

- The paper includes thorough ablation studies and detailed analyses to validate the contributions of each component in the model design.

**Weaknesses:**

- It would be helpful to compare the proposed method with simpler prompting-based approaches such as *Self-Refine* or *Reflexion*, which also aim to improve reasoning through iterative refinement.

- The experiments are limited to Proximal Policy Optimization (PPO). It would strengthen the paper to evaluate how the approach performs with more advanced or stabilized RL algorithms such as Gradient-Regularized Policy Optimization (GRPO).

- In Figure 4(a), there appears to be some overfitting in the baseline methods. Additionally, the performance fluctuation of the proposed method after 1k training steps raises concerns about stability. It is unclear whether the observed improvement is statistically significant or due to noise.

- While I am not deeply familiar with all baselines in this area, I believe there are additional recent works that apply RL to enhance LLM reasoning. A broader baseline comparison would help clarify the significance of the proposed approach.

---

> ### Author Rebuttal · Authors · 2025-07-31
>
> We thank the reviewer for their constructive feedback and detailed comments. We address the reviewer's concerns below.
>
> **Response to Weakness 1&4: Additional Baselines**
>
> We agree that comparing against additional baselines is valuable. We have identified several concurrent works—ThinkPrune [1], ConciseRL [2], Structured Reasoning [3], and AdaptThink [4]—that also apply RL to improve mathematical reasoning on R1.5B.
>
> *Comparison with Concurrent Works Finetuning R1.5B Models*
>
> | Method                            | MATH500 Accuracy | MATH500 Length | AIME24 Accuracy | AIME24 Length | AMC23 Accuracy | AMC23 Length |
> | --------------------------------- | ---------------- | -------------- | --------------- | ------------- | -------------- | ------------ |
> | ThinkPrune [1]                    | 83.2             | 1938           | 27.1            | 5631          | 73.2           | 3039         |
> | ConciseRL [2]                     | 81.0             | 1965           | 30.0            | 6752          | 69.4           | 2936         |
> | SR-FLOW [3]                       | 85.3             | n.a.           | 36.7            | n.a.          | 77.8           | n.a.         |
> | AdaptThink [4]                    | 82.0             | 1782           | 31.0            | 6679          | n.a.           | n.a.         |
> | Baseline                          | 86.2             | 2780           | 35.4            | 5778          | 72.8           | 3938         |
> | Thinker                           | 87.0             | 4061           | 35.6            | 8148          | 81.7           | 5709         |
> | Thinker-Fast                      | 77.2             | 649            | 11.5            | 960           | 59.2           | 833          |
> | Thinker (short verification)      | **88.5**         | 2501           | **39.0**        | 5597          | **83.6**       | 3517         |
> | Thinker-Fast (short verification) | 80.9             | 600            | 18.1            | 853           | 66.9           | 751          |
>
> The table above compares our method against these new baselines. To explore efficiency, we also trained a variant model using a shorter verification step (2k tokens instead of 6k). We present results for this "short verification" model in both its full (`Thinker (short verification)`) and fast (`Thinker-Fast (short verification)`) deployment modes.
>
> Several observations can be made from the results above:
>
> 1. Shortening the verification tokens from 6k to 2k has little impact on final accuracy—showing a slight increase for the R1.5B model. However, the total token length can be significantly reduced in full deployment mode.
> 2. Thinker (short verification) R1.5B models outperform other concurrent works in terms of accuracy while remaining competitive in terms of token length.
> 3. Thinker-Fast (short verification) R1.5B models use significantly fewer tokens than all concurrent works, while maintaining slightly worse performance on simpler benchmarks (MATH500 and AMC23), though with significantly worse performance on complex tasks (AIME24).
>
> An additional advantage of Thinker—beyond the metrics shown above—is that users can choose the deployment mode based on task demands and computational cost constraints (e.g., fast mode for simpler tasks and full mode for complex ones) without training two separate models. In summary, Thinker offers a flexible trade-off while maintaining competitive or even superior performance compared to similar methods.
>
> The above new results on short verification and the additional baselines will be included in the revised version of the main paper.
>
> Regarding simpler prompting-based methods like Self-Refine or Reflexion, we believe a direct comparison is challenging. First, while the effectiveness of these methods has been demonstrated on large, instruction-tuned models like GPT-4, they would likely perform poorly on our smaller base models, which are not instruction-tuned. Second, a direct comparison of the final outputs would be misleading due to the vast difference in computational cost: these prompting strategies are inference-time techniques requiring no finetuning, while our methods and baselines involve substantial RL training. One might suggest resolving these issues by applying RL to these prompting methods; however, this requires designing a good reward signal, which is a significant research topic in its own right. For these reasons, we consider this comparison outside the scope of our current work.
>
> **Response to Weakness 2: GRPO vs. PPO**
>
> We chose PPO as our primary RL algorithm because prior work [5, 6] showed that GRPO can be unstable for finetuning LLMs, sometimes leading to performance collapse. We also conducted preliminary experiments finetuning our models on the standard QA task using GRPO and confirmed the instability analyzed in [5, 6]. Given these findings, we focused our efforts on PPO, which provided more stable and reliable results.
>
> **Response to Weakness 3: Overfitting in the Baseline**
>
> We select the final model for evaluation based on its peak performance on a held-out validation set (i.e., early stopping). For example, the results for the Qwen 1.5B baseline reported in Table 1 are from the checkpoint at training step 900, before the performance degradation occurred.
>
> Regarding statistical significance, we have conducted new multi-seed experiments to assess the variance from RL training. Please refer to our response to Reviewer hUT1 for a detailed presentation of these new results.
>
> **Response to Question 1: Thinker-Fast performing better on GPQA**
>
> We believe this interesting phenomenon is due to GPQA's nature as a knowledge-intensive science benchmark. Unlike our other benchmarks which focus on mathematical reasoning, GPQA is knowledge-intensive, covering biology, physics, and chemistry. Our primary hypothesis is that the fast thinking step helps to mitigate hallucinations on such tasks. This hypothesis is guided by prior work [7], which suggests that long reasoning chains trained via RL can lead to a higher degree of hallucination. While the long thinking process used by the full Thinker model and our baselines is beneficial for math, it may become a vulnerability on a knowledge-based benchmark like GPQA. By explicitly constraining the generation to a short format, Thinker-Fast limits the model's opportunity for flawed reasoning and forces it to rely more directly on its pre-trained knowledge.
>
> This also explains why Thinker-Fast may outperform the full Thinker mode on this specific task. The deployment of the subsequent verification and slow thinking steps in the full mode, while allowing more room for reasoning, appears to partially negate the anti-hallucination benefit of the fast thinking step. The advantage is not entirely lost, however, as the full Thinker mode still outperforms the baseline on GPQA. We are actively exploring the implications of this finding for controlling hallucination in RL-finetuned LLMs.
>
> We thank the reviewer again for their valuable feedback. We believe these new baseline comparisons, along with our clarifications on the training process and algorithm choices, substantially strengthen the paper. We would be grateful if the reviewer would consider this new evidence and our responses in their final assessment. We welcome any further questions.
>
> [1] Hou, B., Zhang, Y., Ji, J., Liu, Y., Qian, K., Andreas, J., & Chang, S. (2025). Thinkprune: Pruning long chain-of-thought of llms via reinforcement learning. *arXiv preprint arXiv:2504.01296*.
>
> [2] Dumitru, R. G., Peteleaza, D., Yadav, V., & Pan, L. (2025). ConciseRL: Conciseness-Guided Reinforcement Learning for Efficient Reasoning Models. *arXiv preprint arXiv:2505.17250*.
>
> [3] Dong, Y., & Fan, H. (2025). Enhancing Large Language Models through Structured Reasoning. *arXiv preprint arXiv:2506.20241*.
>
> [4] Zhang, J., Lin, N., Hou, L., Feng, L., & Li, J. (2025). Adaptthink: Reasoning models can learn when to think. *arXiv preprint arXiv:2505.13417*.
>
> [5] Hu, J., Zhang, Y., Han, Q., Jiang, D., Zhang, X., & Shum, H. Y. (2025). Open-reasoner-zero: An open source approach to scaling up reinforcement learning on the base model. *arXiv preprint arXiv:2503.24290*.
>
> [6] Rajani, N., Gema, A. P., Goldfarb-Tarrant, S., & Titov, I. (2025). Scalpel vs. Hammer: GRPO Amplifies Existing Capabilities, SFT Replaces Them. *arXiv preprint arXiv:2507.10616*.
>
> [7] Li, J., & Ng, H. T. (2025). The Hallucination Dilemma: Factuality-Aware Reinforcement Learning for Large Reasoning Models. *arXiv preprint arXiv:2505.24630.*

---

> ### Comment · Reviewer_5wyw · 2025-08-04
>
> Thanks the authors for the responses. I do not have any further questions and keep my original score.

---

### Official Review · Reviewer_1C42 · 2025-07-02

**Clarity:** 4
**Significance:** 3
**Originality:** 3
**Rating:** 4
**Confidence:** 5

**Summary:**

Inspired by the Dual Process Theory, the paper proposes an approach to train a model to infer using two modes of thinking (fast and slow) in the question-answering setting. Specifically, the model is trained to take a 4-step variant of CoT paradigm, involving a fast thinking as the first step which predicts the answer within a short output token budget. The second verification step check the answer and proceeds to the slow thinking step if the fast thinking answer is verified to be incorrect. The slow thinking step involves a longer output token budget. The final summarization step provides the final answer. Experiments were conducted on an array of math reasoning QA benchmarks to show the superior performance of proposed approach compared to baseline methods. Further analyses were conducted to understand the effect of training on response length and model’s reflection patterns.

**Questions:**

The proposed approach seems to have two tries in answering a question (one from fast and slow if fast thinking is verified as incorrect). However, most baseline methods would only have one try in answering the questions. It seems it is fairer to compare with pass@2 metrics of baseline methods where there are two tries to answer a particular question.

**Ethical Concerns:**

["NO or VERY MINOR ethics concerns only"]

**Final Justification:**

The authors have answered my questions and increased my confidence of the evaluation.

**Limitations:**

yes

**Quality:**

3

**Strengths And Weaknesses:**

Strengths:
Proposed approach is shown to improve math reasoning performance while reducing compute with the fast thinking mode.
The paper is well written and easy to follow

Weaknesses:
Experiments are limited to math reasoning tasks, applicability to other domains and tasks is yet to be supported by experiments

---

> ### Author Rebuttal · Authors · 2025-07-31
>
> We thank the reviewer for their constructive feedback and detailed comments. We address the reviewer's concerns below.
>
> **Response to Weakness 1: Limitation to Math Reasoning Tasks**
>
> The primary focus of this paper is on improving the math reasoning capabilities of LLMs via RL, which is a highly active and focused research area. As evidence of this focus within the community, a recent survey [1] lists multiple concurrent papers dedicated specifically to math reasoning. We are hopeful that our work can contribute to this research frontier.
>
> Nonetheless, we agree that demonstrating broader applicability is valuable. We have performed an evaluation of our models on a non-mathematical benchmark, GPQA Diamond, which consists of 448 challenging multiple-choice questions written by domain experts in biology, physics, and chemistry [2]. Despite being trained exclusively on math questions, our Thinker models also perform well on GPQA Diamond compared to both the baseline and the original pre-trained models (Table 1 in the main paper). This promising generalization performance leaves room for further exploration in future work, which we will acknowledge in the paper.
>
> **Response to Question 1: Two Tries Per Question**
>
> Our Thinker agent does not get two attempts at providing a final answer during inference. The process, as depicted in Figure 2 of the main paper, is as follows:
>
> 1. The model generates a fast answer in the fast thinking step.
> 2. The model verifies if the fast answer is correct in the verification step. This is a step internal to the model's own reasoning process and does not rely on any external verifier.
> 3. If the model verifies the fast answer as correct in the verification step, then the fast answer is defined as the final answer. Otherwise, the model goes into the slow thinking step, and the final answer will be the slow answer given in the slow thinking step.
>
> Crucially, this process results in only one final answer being produced for each question. Therefore, the comparison with baselines is fair, as both our method and the baselines submit only a single, final answer for each question (i.e., the evaluation is Pass@1 for all methods). We will explicitly update the experimental setup section in our paper to make this single-attempt evaluation process clearer.
>
> We thank the reviewer again for their valuable feedback. We hope this clarification addresses the main concern regarding evaluation fairness. We believe that with this resolved, the strengths of the paper as noted by the reviewer—its strong empirical performance and novel approach—solidify its contribution. We would be grateful if the reviewer would consider this clarification in their final assessment and are happy to answer any further questions.
>
> [1] Li, Z. Z., Zhang, D., Zhang, M. L., Zhang, J., Liu, Z., Yao, Y., ... & Liu, C. L. (2025). From system 1 to system 2: A survey of reasoning large language models. *arXiv preprint arXiv:2502.17419.*
>
> [2] Rein, D., Hou, B. L., Stickland, A. C., Petty, J., Pang, R. Y., Dirani, J., ... & Bowman, S. R. (2024, July). Gpqa: A graduate-level google-proof q&a benchmark. *In First Conference on Language Modeling.*

---

> > ### Author Response · Authors · 2025-08-05
> >
> > Dear Reviewer 1C42,
> >
> > As the discussion phase deadline approaches, we would greatly appreciate any additional feedback you might have on our paper or our previous response. We sincerely thank you for the time you've already dedicated to reviewing our work, and we remain available to address any further questions or concerns.
> >
> > Best regards,
> >
> > Authors

---

> > > ### Comment · Reviewer_1C42 · 2025-08-06
> > > **Thank you for the response**
> > >
> > > The reviewer thanks the authors for the response. I have no more questions.

---

### Official Review · Reviewer_fcXk · 2025-07-02

**Clarity:** 3
**Significance:** 3
**Originality:** 2
**Rating:** 4
**Confidence:** 3

**Summary:**

This paper proposes the "Thinker" task, a four-stage RL framework for LLMs (Fast Thinking, Verification, Slow Thinking, Summarization) inspired by Dual Process Theory. By decomposing QA into structured stages with distinct rewards, the method aims to enhance intuition, evaluation, refinement, and integration capabilities. Experiments on Qwen2.5-1.5B and DeepSeek-R1-Distill-Qwen-1.5B show significant accuracy gains across math benchmarks, with Fast Thinking achieving efficiency via strict token budgets. The approach reduces redundant self-reflection but yields longer responses due to verification iterations.

**Questions:**

Why does Thinker achieve higher accuracy despite generating longer responses than standard QA? For instance, DeepSeek-R1-Distill-Qwen-1.5B’s responses increase in token length, yet MATH500 accuracy rises by 8.5%. Is this due to Verification’s iterative self-checks or Slow Thinking’s deep refinement?

**Ethical Concerns:**

["NO or VERY MINOR ethics concerns only"]

**Final Justification:**

The author addressed my concerns, especially the performance improvement on short data in the long context setting. I have adjusted my score accordingly.

**Limitations:**

Yes.

**Paper Formatting Concerns:**

No.

**Quality:**

2

**Strengths And Weaknesses:**

# Strength

1. Inspired by human cognitive systems (System 1/2), Thinker explicitly trains intuition (Fast Thinking) and deliberation (Slow Thinking), forming a virtuous loop where fast responses guide slow refinement and vice versa. This structured design aligns with psychological theories, providing a principled RL framework.

2. Each stage’s reward (e.g., binary for fast thinking, weighted for verification) targets specific capabilities, enabling precise credit assignment.

# Weaknesses

1. In Section 5.3, the authors observe that although the Thinker task exhibits fewer reflection patterns, the verification step nonetheless produces longer responses. This phenomenon stems primarily from repeated self-verification cycles incentivized by the answer extraction strategy, which focuses solely on the final boxed output. This indicates a redistribution of inefficiency rather than a net reduction, potentially compromising practical inference gains in certain scenarios.

---

> ### Author Rebuttal · Authors · 2025-07-31
>
> We thank the reviewer for their constructive feedback and detailed comments. We address the reviewer's concerns below.
>
> **Response to Weakness 1: Inefficiency During Inference**
>
> We acknowledge that one limitation of our method is that longer verification step is unavoidable during inference in full deployment mode (note: in fast deployment mode, the verification step is not needed though). This can indeed lead to longer response lengths for more complex tasks. To mitigate this issue, we conducted an additional experiment on R1.5B and R7B models by shortening the token budget in the verification step from 6k tokens to 2k tokens, and found that performance can be maintained while significantly reducing the number of tokens used. We present results for this "short verification" model in both its full (`Thinker (short verification)`) and fast (`Thinker-Fast (short verification)`) deployment modes:
>
> *Average Accuracy for Finetuned R1.5B Models*
>
> | Method | MATH500 | AIME24 | AIME2025 | GPQA Diamond | Olympiad Bench | AMC23 | Minerva Math | College Math | Average |
> | --------------------------------- | --------- | --------- | --------- | ------------ | -------------- | --------- | ------------ | ------------ | --------- |
> | Baseline | 86.24 | 35.42 | 23.75 | 25.69 | 49.22 | 72.81 | 32.08 | 42.02 | 45.90 |
> | Thinker | 87.02 | 35.62 | **27.71** | 36.08 | 54.21 | 81.72 | 33.23 | **42.77** | 49.80 |
> | Thinker-Fast | 77.21 | 11.46 | 11.46 | 30.08 | 40.39 | 59.22 | 29.23 | 41.37 | 37.55 |
> | Thinker (Short Verification) | **88.51** | **38.96** | 26.67 | **37.41** | **55.49** | **83.59** | **34.77** | 42.44 | **50.98** |
> | Thinker-Fast (Short Verification) | 80.90 | 18.12 | 12.71 | 28.69 | 44.38 | 66.88 | 31.20 | 41.57 | 40.56 |
>
> *Response Length (in Token) for Finetuned R1.5B Models*
>
> | Method | MATH500 | AIME24 | AIME2025 | GPQA Diamond | Olympiad Bench | AMC23 | Minerva Math | College Math | Average |
> | --------------------------------- | ------- | ------ | -------- | ------------ | -------------- | ------ | ------------ | ------------ | ------- |
> | Baseline | 2780.1 | 5777.7 | 5694.1 | 3815.9 | 4369.3 | 3938.2 | 3505.2 | 2769.0 | 4081.2 |
> | Thinker | 4061.0 | 8147.8 | 7819.5 | 6902.6 | 6128.0 | 5708.9 | 5883.7 | 3861.0 | 6064.1 |
> | Thinker-Fast | 648.5 | 960.2 | 919.9 | 732.8 | 843.7 | 833.1 | 720.7 | 616.8 | 784.5 |
> | Thinker (Short Verification) | 2501.0 | 5597.4 | 5355.7 | 5008.4 | 4060.8 | 3517.1 | 3976.7 | 2445.9 | 4057.9 |
> | Thinker-Fast (Short Verification) | 600.3 | 853.1 | 813.2 | 726.2 | 745.6 | 751.0 | 709.8 | 559.9 | 719.9 |
>
> *Average Accuracy for Finetuned R7B Models*
>
> | Method | MATH500 | AIME24 | AIME2025 | GPQA Diamond | Olympiad Bench | AMC23 | Minerva Math | College Math | Average |
> | --------------------------------- | --------- | --------- | --------- | ------------ | -------------- | --------- | ------------ | ------------ | --------- |
> | Baseline | 91.03 | 47.50 | 34.58 | 34.63 | 56.76 | 87.81 | 40.23 | 42.71 | 54.41 |
> | Thinker | **93.04** | 56.25 | **41.46** | **41.51** | **62.12** | **91.09** | **44.39** | **43.29** | **59.14** |
> | Thinker-Fast | 86.48 | 26.46 | 21.88 | 34.12 | 51.77 | 71.56 | 43.08 | 42.58 | 47.24 |
> | Thinker (Short Verification) | 92.16 | **56.46** | 36.67 | 41.07 | 61.80 | 88.91 | 44.26 | 42.91 | 58.03 |
> | Thinker-Fast (Short Verification) | 86.49 | 27.08 | 22.71 | 33.05 | 51.30 | 72.66 | 42.10 | 42.24 | 47.20 |
>
> *Response Length (in Token) for Finetuned R7B Models*
>
> | Method | MATH500 | AIME24 | AIME2025 | GPQA Diamond | Olympiad Bench | AMC23 | Minerva Math | College Math | Average |
> | --------------------------------- | ------- | ------ | -------- | ------------ | -------------- | ------ | ------------ | ------------ | ------- |
> | Baseline | 2313.3 | 5517.6 | 5569.4 | 3567.7 | 3896.0 | 3453.9 | 2950.3 | 2224.5 | 3686.6 |
> | Thinker | 2366.3 | 6736.2 | 6640.3 | 5416.0 | 4195.3 | 3535.9 | 3756.3 | 2171.0 | 4352.2 |
> | Thinker-Fast | 502.0 | 766.8 | 734.1 | 681.5 | 640.4 | 626.2 | 599.0 | 516.9 | 633.4 |
> | Thinker (Short Verification) | 1504.7 | 4495.2 | 4601.0 | 3903.5 | 2762.4 | 2321.8 | 2274.3 | 1398.0 | 2907.6 |
> | Thinker-Fast (Short Verification) | 500.2 | 800.3 | 746.5 | 662.8 | 645.9 | 638.7 | 592.4 | 504.3 | 636.4 |
>
> We have also identified several concurrent works—ThinkPrune [1], ConciseRL [2], Structured Reasoning [3], and AdaptThink [4]—that also apply RL to improve mathematical reasoning on R1.5B with more advanced methods:
>
> *Comparison with Concurrent Works Finetuning R1.5B Models*
>
> | Method | MATH500 Accuracy | MATH500 Length | AIME24 Accuracy | AIME24 Length | AMC23 Accuracy | AMC23 Length |
> | --------------------------------- | ---------------- | -------------- | --------------- | ------------- | -------------- | ------------ |
> | ThinkPrune [1] | 83.2 | 1938 | 27.1 | 5631 | 73.2 | 3039 |
> | ConciseRL [2] | 81.0 | 1965 | 30.0 | 6752 | 69.4 | 2936 |
> | SR-FLOW [3] | 85.3 | n.a. | 36.7 | n.a. | 77.8 | n.a. |
> | AdaptThink [4] | 82.0 | 1782 | 31.0 | 6679 | n.a. | n.a. |
> | Baseline | 86.2 | 2780 | 35.4 | 5778 | 72.8 | 3938 |
> | Thinker | 87.0 | 4061 | 35.6 | 8148 | 81.7 | 5709 |
> | Thinker-Fast | 77.2 | 649 | 11.5 | 960 | 59.2 | 833 |
> | Thinker (short verification) | **88.5** | 2501 | **39.0** | 5597 | **83.6** | 3517 |
> | Thinker-Fast (short verification) | 80.9 | 600 | 18.1 | 853 | 66.9 | 751 |
>
> Several observations can be made from the results above:
>
> 1. Shortening the verification tokens from 6k to 2k has little impact on final accuracy—showing a slight increase for the R1.5B model and a slight decrease for the R7B model. However, the total token length can be significantly reduced in full deployment mode.
> 2. Thinker (short verification) R1.5B models outperform other concurrent works in terms of accuracy while remaining competitive in terms of token length.
> 3. Thinker-Fast (short verification) R1.5B models use significantly fewer tokens than all concurrent works, while maintaining slightly worse performance on simpler benchmarks (MATH500 and AMC23), though with significantly worse performance on complex tasks (AIME24).
>
> An additional advantage of Thinker—beyond the metrics shown above—is that users can choose the deployment mode based on task demands and computational cost constraints (e.g., fast mode for simpler tasks and full mode for complex ones) without training two separate models. In summary, Thinker offers a flexible trade-off while maintaining competitive or even superior performance compared to similar methods.
>
> The above new results on short verification and the additional baselines will be included in the revised version of the main paper.
>
> **Response to Question 1: Thinker achieving higher accuracy despite generating longer responses than standard QA**
>
> In general, longer responses are usually associated with higher accuracy, as the model has more tokens to perform reasoning. The DeepSeek R1 paper [5] shows that both response length and accuracy increase during training. The increase in token length during training for both the baseline and Thinker models, as shown in Fig. 5b, is consistent with this trend. Manual inspection reveals that the models generally learn to perform more self-correction during training—e.g., "Wait, I may be wrong here, let me recheck"—for both the baseline and Thinker models. This behavior aligns with the final reward being based on the correctness of the final answer, giving the agent an incentive to perform more verification and try additional approaches when there is room to use more tokens.
>
> We thank the reviewer again for their valuable feedback. We believe these new baseline comparisons, along with our new results, substantially strengthen the paper. We would be grateful if the reviewer would consider this new evidence and our responses in their final assessment. We welcome any further questions.
>
> [1] Hou, B., Zhang, Y., Ji, J., Liu, Y., Qian, K., Andreas, J., & Chang, S. (2025). Thinkprune: Pruning long chain-of-thought of llms via reinforcement learning. *arXiv preprint arXiv:2504.01296*.
>
> [2] Dumitru, R. G., Peteleaza, D., Yadav, V., & Pan, L. (2025). ConciseRL: Conciseness-Guided Reinforcement Learning for Efficient Reasoning Models. *arXiv preprint arXiv:2505.17250*.
>
> [3] Dong, Y., & Fan, H. (2025). Enhancing Large Language Models through Structured Reasoning. *arXiv preprint arXiv:2506.20241*.
>
> [4] Zhang, J., Lin, N., Hou, L., Feng, L., & Li, J. (2025). Adaptthink: Reasoning models can learn when to think. *arXiv preprint arXiv:2505.13417*.
>
> [5] Guo, D., Yang, D., Zhang, H., Song, J., Zhang, R., Xu, R., ... & He, Y. (2025). Deepseek-r1: Incentivizing reasoning capability in llms via reinforcement learning. *arXiv preprint arXiv:2501.12948*.

---

> > ### Author Response · Authors · 2025-08-05
> >
> > Dear Reviewer fcXk,
> >
> > As the discussion phase deadline approaches, we would greatly appreciate any additional feedback you might have on our paper or our previous response. We sincerely thank you for the time you've already dedicated to reviewing our work, and we remain available to address any further questions or concerns.
> >
> > Best regards,
> >
> > Authors

---

> > ### Comment · Reviewer_fcXk · 2025-08-07
> >
> > Thanks for your reply, I increase my score to 4.

---

### Official Review · Reviewer_hUT1 · 2025-07-02

**Clarity:** 3
**Significance:** 3
**Originality:** 2
**Rating:** 4
**Confidence:** 3

**Summary:**

The paper presents "Thinker", a method of decomposing reasoning tasks into four subtasks, and associated RL training rewards for each subtask. Motivated by Dual Process Theory for human cognition, the subtasks are "fast thinking", "verification", "slow thinking", and summarization.

Fast thinking consists of attempting to answer the question with a small token budget (1k tokens). Verification consists of using a large token budget to assess the answer produced by fast thinking. If verification rejects the fast thinking result, slow thinking uses a large token budget to produce a corrected answer. Finally, summarization produces a new short reasoning trace, trained with a reward for producing reasoning which both achieves the same answer as slow thinking, and is assigned high probability under the fast thinking prompt.

Models are trained via PPO to solve math questions. It studies two models (Qwen2.5 1.5B, and DeepSeek-R1-Distill-Qwen-1.5B) and two task settings (“baseline”, i.e. standard QA reward, vs. Thinker subtasks). Thinker is shown to result in larger performance gains than the direct QA task: on Qwen, Thinker improves performance by 24%, while Baseline improves performance by 21%; and on R1, Thinker improves performance by 14%, while Baseline improves performance by 10%.

**Questions:**

How are the standard errors in appendix B computed? E.g. for the MATH 500 Thinker: if rewards are binary correctness, for the Thinker results, (64% accuracy), I get sqrt(.64\*(1-.64)/500) ~= 2.1%. If we considered each of the 500\*16 samples as IID, we’d get sqrt(.64\*(1-.64)/(500\*16)) ~= 0.54%; this would still be higher than reported in the paper, though note that this will underestimate standard errors due correlation between samples on the same question (e.g. see “cluster standard errors”: https://www.anthropic.com/research/statistical-approach-to-model-evals)

I searched the code for “error” and “std”, but it wasn’t clear to me where these statistics are computed vs. using std for internal model normalization.

**Ethical Concerns:**

["NO or VERY MINOR ethics concerns only"]

**Final Justification:**

Overall rating updated 2->4, quality 2->3, significance 2->3. The authors' additional experiments and analysis provide good evidence that their results are not merely due to chance:

1) They performed one of their RL runs with 3 independent seeds, demonstrating that the variation between seeds is small compared to the variation between runs. This addressed my most critical concern, that the headline results could be simply due to randomness. (While it would be preferable to have multiple seeds for each parameter setting, in order to rule out heteroskedasticiy, I don't have a strong reason to expect heteroskedasticity a priori.)
2) They conducted more thorough statistical analysis, showing on which datasets improvement is significant (treating dataset examples as the random variable).

As such, I believe this paper provides value with its approach for improving model reasoning.

**Limitations:**

Yes

**Quality:**

3

**Strengths And Weaknesses:**

**Strengths:**

1) The paper is clearly presented, and the method is described well.
2) Evaluations are performed on 7 datasets, consisting of 6 math benchmarks plus GPQA Diamond.
3) The paper presents some interesting qualitative analysis; I found it interesting that “reflection patterns” seem to increase over training for both Baseline and Thinker, despite much more flat total response length.

**Weaknesses**:

I’m concerned about how much we can infer from the presented results. IIUC the paper's conclusions stem from 5 total PPO RL runs:
* R1 1.5B, Thinker vs Baseline
* Qwen 1.5B, Thinker vs. Baseline
* Qwen 1.5B, Thinker without Summarization

The paper does provide CIs with respect to dataset examples (though for readability these should ideally be included in the main table, rather than separately in the appendix). However, with only single RL runs for each setting, we have no way of estimating variation due to RL training dynamics. E.g. figure 6 shows divergence in “reflection pattern occurrence”; is this actually caused by differences in the loss, or simply that Thinker without Summarization happened by chance to produce some high-quality reflection patterns, and so settled into a different but equally effective writing style?

The average performance improvements over the baseline are relatively minor (3% and 4%); to adequately support the paper's claim and justify the substantial increase in method complexity, the paper should compare multiple RL runs for each setting (each using different seeds for dataset shuffling and LLM sampling), and show statistical significance between baseline and thinker with respect to variation in RL runs.

---

> ### Author Rebuttal · Authors · 2025-07-31
>
> We thank the reviewer for their constructive feedback and detailed comments. We address the reviewer's concerns below.
>
> **Response to Weakness 1: Variance of Performance and Number of Experiments**
>
> *New Multi-Seed Experiments*
>
> We agree with the reviewer that assessing the variance from RL training dynamics is important for validating the benefits of the proposed method. To address this concern, we have conducted new experiments for our main result (Q1.5B Thinker vs. Baseline), running each method with three different random seeds. The aggregated results are as follows:
>
> | Method            | MATH500 | AIME24 | AIME2025 | GPQA Diamond | Olympiad Bench | AMC23 | Minerva Math | College Math | Average |
> | ----------------- | ------- | ------ | -------- | ------------ | -------------- | ----- | ------------ | ------------ | ------- |
> | Baseline (Seed 1) | 57.98   | 3.33   | 3.33     | 21.46        | 24.54          | 34.38 | 17.78        | 36.21        | 24.88   |
> | Baseline (Seed 2) | 59.26   | 4.58   | 1.46     | 19.57        | 25.05          | 35.62 | 18.43        | 38.40        | 25.30   |
> | Baseline (Seed 3) | 58.44   | 2.92   | 1.04     | 19.73        | 26.52          | 35.47 | 18.84        | 38.39        | 25.17   |
> | ORZ [1]           | 58.00   | 3.50   | 1.00     | 16.80        | -              | -     | -            | -            | -       |
> | SimpleRL [2]      | 59.00   | 4.20   | -        | -            | 21.00          | 35.00 | 20.20        | -            | -       |
> | RL on DSR [3]     | 57.20   | 5.00   | 0.80     | -            | 21.20          | 30.30 | 17.60        | -            | -       |
> | Thinker (Seed 1) | 64.25 | 6.25 | 2.50 |  23.74 | 28.11 | 40.62 | 19.03 | 38.33 | 27.85 |
> | Thinker (Seed 2) | 64.53 | 6.88 | 2.29 | 19.54 | 28.03 | 39.06 | 19.44 | 38.75 | 27.31 |
> | Thinker (Seed 3) | 63.76 | 5.83 | 0.83 | 21.31 | 27.80 | 40.31 | 18.75 | 37.94 | 27.07 |
>
> We note that our baseline results are consistent with those reported by similar methods like ORZ [1], SimpleRL [2], and RL on DSR [3], which all performed RL training on the same Q1.5B model with a standard QA task, similar to our baseline. From the above table, we observe Thinker consistently outperforms the baselines. An independent samples t-test showed that Thinker (M = 27.41, SD = 0.40) performed statistically significantly better than the baseline (M = 25.12, SD = 0.22), t(4) = 8.76, p < .001. Additional comparisons between Thinker and four concurrent works with more advanced methods are provided in our response to Reviewer fcXk.
>
> To address the reviewer's question about the "reflection pattern occurrence" (Figure 6), we analyzed this metric across our new multi-seed runs. The average scores, both over all training steps (steps 0-1200) and during the final training steps (steps 1100-1200), are presented below:
>
> | Horizon              | Thinker (Seed 1) | Thinker (Seed 2) | Thinker (Seed 3) | Thinker without Summarization |
> | -------------------- | ---------------- | ---------------- | ---------------- | ----------------------------- |
> | All Training Steps   | 0.01380          | 0.01686          | 0.01774          | 0.03031                       |
> | Final Training Steps | 0.00897          | 0.02643          | 0.01116          | 0.03124                       |
>
> Across all three seeds, we observe that Thinker models consistently exhibit lower reflection pattern scores compared to the variant without the summarization subtask. This supports our hypothesis that the summarization step encourages the agent to develop a more concise chain-of-thought, thereby reducing reflective or repetitive language. However, as noted by the reviewer, it remains possible that the `Thinker without Summarization` may have converged to a different local optimum with inherently lower reflection patterns. Addressing this possibility would require additional seeds for `Thinker without Summarization`, which are currently beyond our computational budget. Nonetheless, the results presented in the table indicate that Thinker is less likely to settle into policies characterized by high reflection patterns, which we believe sufficiently demonstrates the benefits of incorporating the summarization step.
>
> *Additional Experiments*
>
> At the time of submission, the R7B runs were ongoing due to computational constraints, which is why their preliminary results were placed in the appendix. We have since completed these experiments and present the full results below:
>
> | Method       | MATH500   | AIME24    | AIME2025  | GPQA Diamond | Olympiad Bench | AMC23     | Minerva Math | College Math | Average   |
> | ------------ | --------- | --------- | --------- | ------------ | -------------- | --------- | ------------ | ------------ | --------- |
> | Pretrained   | 84.05     | 37.50     | 28.54     | 17.58        | 37.92          | 36.41     | 34.49        | 40.72        | 39.65     |
> | Baseline     | 91.03     | 47.50     | 34.58     | 34.63        | 56.76          | 87.81     | 40.23        | 42.71        | 54.41     |
> | Thinker      | **93.04** | **56.25** | **41.46** | **41.51**    | **62.12**      | **91.09** | **44.39**    | **43.29**    | **59.14** |
> | Thinker-Fast | 86.48     | 26.46     | 21.88     | 34.12        | 51.77          | 71.56     | 43.08        | 42.58        | 47.24     |
>
> As shown, Thinker demonstrates consistent performance improvements over the baseline on the R7B model. This result is also consistent with our findings on smaller models and suggests that our approach scales effectively.
>
> Furthermore, to address potential concerns about method efficiency, we have also conducted two additional experiments (detailed in our response to Reviewer fcXk) showing that the token budget for the verification step can be significantly reduced while maintaining performance, thereby improving Thinker's efficiency.
>
> In total, these new experiments bring the number of PPO runs presented to 13:
>
> - Q1.5B: Thinker vs. Baseline (3 seeds per run)
> - Q1.5B: Thinker without summarization
> - R1.5B: Thinker vs. Baseline
> - R1.5B: Thinker with short verification
> - R7B: Thinker vs. Baseline
> - R7B: Thinker with short verification
>
> Each PPO run is computationally intensive, requiring approximately 960 A100-hours (8 GPUs for 5 days). The experiments presented here, including the new runs, represent over 10,000 GPU-hours. Given these substantial costs and our limited computational budget, we have focused our resources on the most important experiments. We hope this comprehensive evaluation is sufficient to address the concerns about statistical robustness.
>
> We will incorporate all new results and analyses into the revised manuscript.
>
> **Response to Question 1: Calculation of Standard Errors**
>
> The reviewer is correct that the method for calculating standard errors was not explicitly defined. We apologize for this oversight.
>
> The standard error reported in Appendix B is not calculated over the questions in the dataset, but rather over 16 independent samples for a given benchmark. The procedure is as follows:
>
> 1. For a single model, we generate 16 samples per question from the benchmark with a temperature of 1.0.
> 2. This process yields 16 aggregate accuracy scores for the benchmark.
> 3. The standard error is then calculated over these 16 accuracy scores.
>
> The purpose of this metric is to quantify the sampling variance of a single trained model (particularly relevant for benchmarks with a limited number of questions, such as AIME), which is distinct from the training variance observed across different RL runs. Our original intent was for this to be conveyed in the caption of Table 1, by placing the reference to the standard error after the sentence: “All scores are Pass@1 accuracy (%) averaged over 16 samples.” However, we now recognize that this implicit connection was too subtle. In the revised paper, we will include a clear and explicit definition of the standard error.
>
> We thank the reviewer again for their valuable feedback. We hope our new experiments sufficiently address the concerns about training variance. We would be grateful if the reviewer would consider this new evidence in their final assessment. We welcome any further questions.
>
> [1] Hu, J., Zhang, Y., Han, Q., Jiang, D., Zhang, X., & Shum, H. Y. (2025). Open-reasoner-zero: An open source approach to scaling up reinforcement learning on the base model. *arXiv preprint arXiv:2503.24290*.
>
> [2] Zeng, W., Huang, Y., Liu, Q., Liu, W., He, K., Ma, Z., & He, J. (2025). Simplerl-zoo: Investigating and taming zero reinforcement learning for open base models in the wild. *arXiv preprint arXiv:2503.18892*.
>
> [3] Wang, Y., Yang, Q., Zeng, Z., Ren, L., Liu, L., Peng, B., ... & Shen, Y. (2025). Reinforcement learning for reasoning in large language models with one training example. *arXiv preprint arXiv:2504.20571*.

---

> > ### Comment · Reviewer_hUT1 · 2025-08-04
> >
> > Thanks for your response.
> >
> > **Weakness 1: Variance of Performance and Number of Experiments**
> >
> > The 3-seed run does a lot to address my concern about the main result. While it would be nice to have at least 3 seeds for all runs, this at least gives some evidence that accuracy is fairly stable, and the random variation in a run is lower than the difference between the methods; and I do understand that RL runs can be quite expensive.
> >
> > Thanks for also sharing the results about reflection pattern occurrence. Unlike the main results, the variation between seeds looks comparable to the variation between groups; I think it's still fine to mention reflection pattern occurrence as an observation, but with the current evidence I'd avoid drawing conclusions based on the observed difference.
> >
> > **Question 1: Calculation of Standard Errors**
> >
> > Thanks for the explanation of your current method.
> >
> > While it may be useful to know that models have little variation within their samples on a given question, typically standard errors are used to answer questions like "is model A actually better than model B on this task, or could the difference have arisen by chance?" What we care about in principle isn't model A's performance on these specific sets of questions; it's model A's performance on the distribution which generated these sets of questions.
> >
> > If we used your approach to evaluate two models with temperature 0, standard errors would be 0; but we couldn't claim to reject the null hypothesis of equal performance with infinite confidence, because maybe the individual questions sampled just happened to play to model A's strengths. (The smaller the dataset, the more important this consideration is.)
> >
> > The right way to handle stochasticity in both model samples and evaluation questions is via clustered errors, as described in https://www.anthropic.com/research/statistical-approach-to-model-evals. Alternatively, you could treat the model's 16-sample average accuracy on a given question as one IID sample, and use a standard statistical test like a bootstrap test (https://docs.scipy.org/doc/scipy/reference/generated/scipy.stats.bootstrap.html) over examples.

---

> > > ### Author Response · Authors · 2025-08-05
> > >
> > > **Weakness 1: Variance of Performance and Number of Experiments**
> > >
> > > Thank you for this feedback. We are glad the 3-seed run helps address the concern regarding the main results, and we appreciate your understanding of the computational expense of these runs.
> > >
> > > We agree that the variation between seeds is comparable to the variation between groups. Accordingly, we will revise our discussion to frame the reflection pattern occurrence as an exploratory observation and explicitly note the variance across seeds.
> > >
> > > **Question 1: Calculation of Standard Errors**
> > >
> > > We are grateful for your guidance and for pointing us to [1]. We agree that this approach provides a more insightful and rigorous method for comparing models than our original approach. We have now adopted the recommended methodology. Specifically:
> > >
> > > 1. For our main performance tables, we calculate the standard error across question-level average scores using Equation (1) from [1], which is the appropriate method for independent questions. The results are presented in the format of Table 2 from the paper.
> > > 2. For the comparison between our Thinker model and the Baseline, we now include a full paired analysis. The standard error of the difference is calculated using Equation (7) from [1], and the results are presented in the format of Table 5 from the paper.
> > >
> > > *Table: Performance of Q1.5B-based models across benchmarks*
> > >
> > > | Benchmark | # Questions | Pretrained | Baseline | Thinker-Fast | Thinker |
> > > |:---|:---|:---|:---|:---|:---|
> > > |MATH 500|500|9.05%(0.63)|57.98%(1.79)|61.60%(1.89)|**64.25%**(1.85)|
> > > |AIME 2024|30|0.00%(0.00)|3.33%(1.58)|**6.25%**(4.10)|**6.25%**(3.92)|
> > > |AIME 2025|30|0.00%(0.00)|**3.33%**(1.79)|2.50%(1.74)|2.50%(1.74)|
> > > |GPQA Diamond|198|4.55%(0.52)|21.46%(1.80)|**26.39%**(2.10)|23.74%(2.04)|
> > > |Olympiad Bench|675|3.09%(0.30)|24.54%(1.32)|24.78%(1.41)|**28.11%**(1.48)|
> > > |AMC23|40|4.06%(1.22)|34.38%(5.62)|35.94%(6.28)|**40.62%**(6.69)|
> > > |Minerva Math|272|2.30%(0.41)|17.78%(1.85)|18.66%(1.92)|**19.03%**(1.94)|
> > > |College Math|2781|7.40%(0.26)|36.22%(0.82)|37.86%(0.86)|**38.34%**(0.87)|
> > >
> > > *Table: Paired Analysis of Thinker Q1.5B v.s. Baseline Q1.5B model*
> > >
> > > | Eval | Model - Baseline | 95% Conf. Interval | Correlation |
> > > |:---|:---|:---|:---|
> > > |MATH 500|**+6.28%(1.02)***|(+4.27,+8.28)|0.84|
> > > |AIME 2024|+2.92%(4.05)|(-5.02,+10.86)|0.12|
> > > |AIME 2025|-0.83%(0.58)|(-1.97,+0.30)|0.95|
> > > |GPQA Diamond|+2.27%(2.13)|(-1.90,+6.45)|0.39|
> > > |Olympiad Bench|**+3.57%(0.83)***|(+1.96,+5.19)|0.83|
> > > |AMC23|+6.25%(3.61)|(-0.82,+13.32)|0.84|
> > > |Minerva Math|+1.24%(1.10)|(-0.91,+3.40)|0.83|
> > > |College Math|**+2.12%(0.39)***|(+1.36,+2.88)|0.90|
> > >
> > > *Table: Performance of R1.5B-based models across benchmarks*
> > >
> > > | Benchmark | # Questions | Pretrained | Baseline | Thinker-Fast | Thinker |
> > > |:-------------|:----------|:------------|:------------|:------------|:----------------|
> > > |MATH 500|500|76.21%(1.51)|86.24%(1.25)|77.21%(1.58)|**87.02%**(1.23)|
> > > |AIME 2024|30|17.50%(5.07)|35.42%(7.09)|11.46%(4.39)|**35.62%**(7.22)|
> > > |AIME 2025|30|17.92%(6.13)|23.75%(6.74)|11.46%(4.70)|**27.71%**(7.38)|
> > > |GPQA Diamond|198|13.76%(1.32)|25.69%(2.00)|30.08%(2.16)|**36.08%**(2.14)|
> > > |Olympiad Bench|675|37.46%(1.59)|49.22%(1.66)|40.39%(1.61)|**54.21%**(1.64)|
> > > |AMC23|40|55.94%(6.35)|72.81%(5.18)|59.22%(6.29)|**81.72%**(4.53)|
> > > |Minerva Math|272|24.82%(2.11)|32.08%(2.47)|29.23%(2.37)|**33.23%**(2.51)|
> > > |College Math|2781|38.87%(0.83)|42.04%(0.89)|41.38%(0.88)|**42.79%**(0.89)|
> > >
> > > *Table: Paired Analysis of Thinker R1.5B v.s. Baseline R1.5B model*
> > >
> > > | Eval | Model - Baseline | 95% Conf. Interval | Correlation |
> > > |:-------------|:------------------|:-----------------|:----------|
> > > |MATH 500|+0.79%(0.59)|(-0.37,+1.95)|0.89|
> > > |AIME 2024|+0.21%(3.64)|(-6.92,+7.34)|0.87|
> > > |AIME 2025|+3.96%(2.71)|(-1.34,+9.26)|0.93|
> > > |GPQA Diamond|**+10.39%(1.77)***|(+6.91,+13.86)|0.64|
> > > |Olympiad Bench|**+4.99%(0.76)***|(+3.51,+6.47)|0.90|
> > > |AMC23|**+8.91%(2.70)***|(+3.61,+14.20)|0.85|
> > > |Minerva Math|+1.15%(1.18)|(-1.16,+3.46)|0.89|
> > > |College Math|**+0.75%(0.31)***|(+0.15,+1.36)|0.94|
> > >
> > > As the new paired analysis shows, our Thinker model achieves statistically significant improvements over the baseline on several key benchmarks. For the remaining benchmarks where the difference is not statistically significant (e.g., AIME 2024 and 2025), we note that the small number of questions limits the statistical power of the test, meaning a very large performance gap would be required to achieve significance.
> > >
> > > Once again, we thank you for your time and valuable feedback. By incorporating multi-seed runs and adopting the rigorous paired analysis, we have worked diligently to address your initial concerns and significantly strengthen the validity of our results. We would be grateful if you would be willing to reconsider your score in light of these improvements.
> > >
> > > [1] Miller, E. (2024). Adding error bars to evals: A statistical approach to language model evaluations. arXiv preprint arXiv:2411.00640.

---

> > > > ### Comment · Reviewer_hUT1 · 2025-08-05
> > > >
> > > > Thank you for performing this statistical analysis. I agree that based on this analysis, many of the results are indeed significant. Please do include these results in the camera ready. My concerns have been addressed and I will update my score.

---

### Comment · Area_Chair_fBki · 2025-08-03
**Post-rebuttal Discussion**

The author has provided a rebuttal to respond to your comments. Please take a look on the author response and discuss with authors if necessary.

Thanks,

AC

---

### Decision · Program_Chairs · 2025-09-17

**Decision:**

Accept (poster)

**Comment:**

**Summary:** This paper proposes Thinker, a reinforcement learning framework for enhancing mathematical reasoning in large language models by explicitly decoupling reasoning into two components: fast thinking (intuitive generation) and slow thinking (deliberation and verification). Inspired by cognitive psychology (System 1 vs. System 2), the approach uses distinct reward structures to guide each stage—binary rewards for fast thinking and weighted verification for slow thinking—thereby enabling more precise credit assignment. Experiments are conducted across six math reasoning benchmarks plus GPQA Diamond, with qualitative analyses of reflection patterns and ablation studies. The results indicate modest but consistent improvements (≈3–4%) over strong baselines, alongside evidence of differentiated reasoning behaviors.

**Strengths:**

- Principled design: The dual-process (fast vs. slow) structure is well motivated by cognitive psychology and provides a systematic RL framework for reasoning.

- Technical novelty: Clear reward decomposition and role assignment across different stages of reasoning.

- Thorough analysis: Includes qualitative analyses (reflection patterns, response length dynamics) and ablation studies to illustrate component contributions.

**Weaknesses:**

- Statistical robustness: As raised by Reviewer hUT1, the main results rely on only single RL runs per setting. Without multiple seeds, it is unclear whether improvements are statistically significant or an artifact of stochastic training dynamics.

- Limited gains: The performance improvements (3–4%) are relatively modest compared to the added complexity of the framework.

- Restricted evaluation: Experiments are limited to math reasoning tasks, leaving generalizability to other domains unexplored.

- Efficiency concerns: While fast thinking is designed to reduce compute, slow thinking often produces longer responses, raising doubts about real inference efficiency.

- Training stability: Some figures (e.g., fluctuations after 1k steps) suggest instability, but this is not deeply analyzed.

My recommendation is acceptence after reading the reviews and the rebuttal.